# A rare variant of African ancestry activates 8q24 lncRNA hub by modulating cancer associated enhancer

Kaivalya Walavalkar [1], Bharath Saravanan[1,2], Anurag Kumar Singh[1], Ranveer Singh Jayani [3], Ashwin Nair[1,2], Umer Farooq[1,4], Zubairul Islam[1,2], Deepanshu Soota[1], Rajat Mann[1], Padubidri V. Shivaprasad [1], Matthew L. Freedman [5,6,7], Radhakrishnan Sabarinathan[1], Christopher A. Haiman[8] & Dimple Notani [1✉]

Genetic variation at the 8q24 locus is linked with the greater susceptibility to prostate cancer in men of African ancestry. One such African ancestry specific rare variant, rs72725854 (A>G/T) (~6% allele frequency) has been associated with a ~2-fold increase in prostate cancer risk. However, the functional relevance of this variant is unknown. Here we show that the variant rs72725854 is present in a prostate cancer-specific enhancer at 8q24 locus. Chromatin-conformation capture and dCas9 mediated enhancer blocking establish a direct regulatory link between this enhancer and lncRNAs PCAT1, PRNCR1 and PVT1. The risk allele ('T') is associated with higher expression of PCAT1, PVT1 and c-myc in prostate tumors. Further, enhancer with the risk allele gains response to androgen stimulation by recruiting the transcription factor SPDEF whereas, non-risk alleles remain non-responsive. Elevated expression of these lncRNAs and c-myc in risk allele carriers may explain their greater susceptibility to prostate cancer.

[1] Genetics and Development, National Centre for Biological Sciences, Tata Institute of Fundamental Research, Bangalore, Karnataka 560065, India. [2] Sastra Deemed University, Thanjavur, Tamil Nadu 613401, India. [3] Howard Hughes Medical Institute, Department of Medicine, University of California San Diego, 9500 Gilman Drive, La Jolla, CA 92037, USA. [4] Trans-Disciplinary University, IVRI road, Bangalore, Tamil Nadu 560064 Karnataka, India. [5] Department of Medical Oncology, Dana-Farber Cancer Institute, Boston, MA 02215, USA. [6] The Eli and Edythe L. Broad Institute, Cambridge, MA 02142, USA. [7] Centre for Functional Cancer Epigenetics, Dana-Farber Cancer Institute, Boston, MA 02215, USA. [8] Department of Preventive Medicine, Keck School of Medicine, University of Southern California, Los Angeles, CA 90007, USA. ✉email: dnotani@ncbs.res.in

The greater incidence of prostate cancer observed in men of African ancestry is due, at least in part, to genetic risk factors[1,2]. Genome-wide association studies (GWAS) have identified more than 180 common variants [minor allele frequency (MAF) >1%], which account for ~35% of the familial risk of prostate cancer in populations of European ancestry. Notably, chromosome 8q24, with ~15 independent risk variants, harbors a disproportionate amount of risk[3]. The majority of these variants are substantially more common in men of African ancestry and may contribute to the greater risk observed in this population[4–6]. Variant rs72725854 at 8q24 is the most significant genetic risk factor for prostate cancer in men of African ancestry[7,8]. The variant is triallelic (A>G/T) with the risk allele ("T") only observed in populations of African ancestry at a frequency of ~6% and found to be associated with a >2-fold increase in prostate cancer risk.

The 8q24 region has been linked with numerous cancer types, however the locus harbors only a few protein-coding genes such as *FAM80B* and the proto-oncogene *MYC* but, several lncRNA genes including, *PCAT1, PCAT2, PRNCR1, CCAT1, CCAT2, CASC19, CASC21,* and *PVT1*. Except for *CASC19* and *21*, all other lncRNAs have been linked to various cancers[9–16]. Apart from c-myc, high levels of PCAT1 and PVT1 have been reported in prostate tumors, suggesting a common mechanism of their transcriptional dysregulation. However, how these lncRNAs are upregulated in these tumors is not clear. Further, if these genes are co-regulated by a common single enhancer remains unknown. Moreover, how African ancestry specific rare risk variants alter these enhancers conferring susceptibility is completely unknown.

Here we show that the 8q24 prostate cancer rare variant, rs72725854 is present in an enhancer and regulates multiple lncRNAs genes namely, PCAT1, PRNCR1, PVT1, and proto-oncogene MYC in the region. We discovered that risk variant of rs72725854 augments the transcriptional activity of the enhancer and sensitizes it to androgen stimulation thereby activating these lncRNAs and c-myc in the region. These findings implicate biological mechanisms through which the rare variant rs72725854 influences prostate cancer risk in men of African ancestry.

## Results

The majority of GWAS risk loci lie in non-coding regions and are enriched for regulatory elements[17]. However, the biological mechanisms for the vast majority are still unclear[9,18–20]. The variant rs72725854 was identified as the most statistically significant SNP in the 8q24 region and genome-wide in men of African ancestry[7,8] and resides in a non-coding region between *PCAT1* and *PCAT2* lncRNAs. To test whether rs72725854 and its linked SNP rs114798100 ($r^2 = 0.8$)[13] has any functional roles, we investigated the presence of androgen receptor (AR) binding using publicly available ChIP-seq data in the prostate cancer cell line LNCaP since, AR modulates gene expression in prostate tumors by binding with distal regulatory elements marked by H3K27ac, H3K4me1, and RNA polymerase II[21]. Interestingly, only the region harboring rs72725854 exhibited an enrichment for these marks suggesting that the region is a potential enhancer (Fig. 1a). Further, to test the regulatory potential of this region and if it is conserved across the cells/tissues of different lineage and their respective tumor cell lines, we interrogated the presence of open chromatin features that are suggestive of regulatory potential by DNase I hypersensitive sites (DHS) available in public domain. Interestingly, only the cancer cells from prostate and liver origin showed a DHS signal around the SNP-harboring region (Fig. 1b) but, between them the prostate cancer cells exhibited higher signal suggesting that this region is likely a prostate cancer-specific enhancer. Further, to understand whether this enhancer is also present in normal prostate epithelial cells, we checked for DHS and H3K27ac marks at this region in immortalized prostate epithelial cells (PrEC) and in LNCaP cells. Interestingly, the DHS and enhancer marks were absent in prostate epithelial cells (Fig. 1c), suggesting that the enhancer is prostate tumor specific. Similarly, FOXA1 is a pioneering factor, which triggers the opening of regulatory regions[22], its binding was seen at the SNP-harboring region in both, prostate tumor samples and in the LNCaP (Fig. 1c). Also, the presence of ATAC-seq signal in this region in several prostate tumors from TCGA cohort supports its regulatory potential as observed in LNCaP cells (Fig. 1d). These data suggest that the SNP-harboring region is likely inactive in healthy prostate epithelial cells and acquires enhancer marks during prostate cancer development.

To test whether the region harboring rs72725854 has regulatory potential in prostate tumors, we performed luciferase reporter assays with major allele "A" in LNCaP cells grown in complete serum (contains hormones and growth factors). The SNP-harboring region exhibited several fold higher activity as compared to the empty plasmid (Fig. 2a), suggesting an active enhancer potential of the rs72725854 region at the 8q24 risk locus in prostate cancer cells. LNCaP cells are homozygous for the non-risk allele (AA) of rs72725854 (Supplementary Fig. 1a), however the region still exhibits enhancer activity suggesting that even the non-risk allele is capable of enhancer function (Fig. 2a). Thus, to understand how the risk allele "T" alters enhancer function, we monitored the allele-specific transcriptional activity of rs72725854. rs72725854 variant is triallelic (A>G/T), with a "G" allele only observed in populations of European ancestry with a frequency of ~2%. We observed the "T" allele to have a ~4-fold increase in reporter activity compared to the "A" or "G" alleles in complete serum (Fig. 2b). This suggests that even though the "A" allele has potential enhancer activity, the risk allele "T" further amplifies the enhancer activity in reporter assays. Interestingly, LNCaP and prostate tumors, but not the healthy tissues, exhibited AR and FOXA1 binding at this region (Fig. 2c), again confirming the specificity of the enhancer to prostate tumors. To test whether the binding of AR to AA genotype in LNCaP allows it to respond to androgens (DHT, agonist of androgens), we tested the relative response of individual alleles to androgens as the liganded AR triggers the transcriptional activation of AR responsive genes in prostate tumors[23]. Toward this, we performed reporter assays on LNCaP cells grown in charcoal stripped serum for three days to remove basal levels of androgens followed by addition of 10 nM DHT or methanol. Interestingly, unlike full serum, where "T" allele exhibited highest activity, the activity of "T" allele dropped to almost similar levels to that of "A" and "G" alleles in stripped media, suggesting that the components that are absent in stripped serum are required for the higher activity of "T" allele (Fig. 2d). Strikingly, the "T" allele gained significant activity in response to added DHT, whereas the "A" and "G" alleles remained unaffected to androgen stimulation (Fig. 2d), suggesting that the higher activity of enhancer with the risk allele "T" (Fig. 2b) is due to its response to androgen. Since the nuclear receptor responsive enhancers exhibit robust induction of eRNAs upon ligand stimulation[24,25], we measured eRNA levels on AR-bound rs72725854 enhancer region upon DHT treatment in LNCaP cells that have the non-risk genotype "A/A" (Supplementary Fig. 1a). The eRNA expression from the enhancer did not change upon DHT stimulation (Fig. 2e). These data confirm that although the region is an active enhancer in prostate tumors and exhibits AR binding but it is non-responsive to androgens. Thus, we hypothesized that AR binding on this region in LNCaP and tumors is indirect either by some other transcription factor or by the virtue of looping with other AR-bound regions.

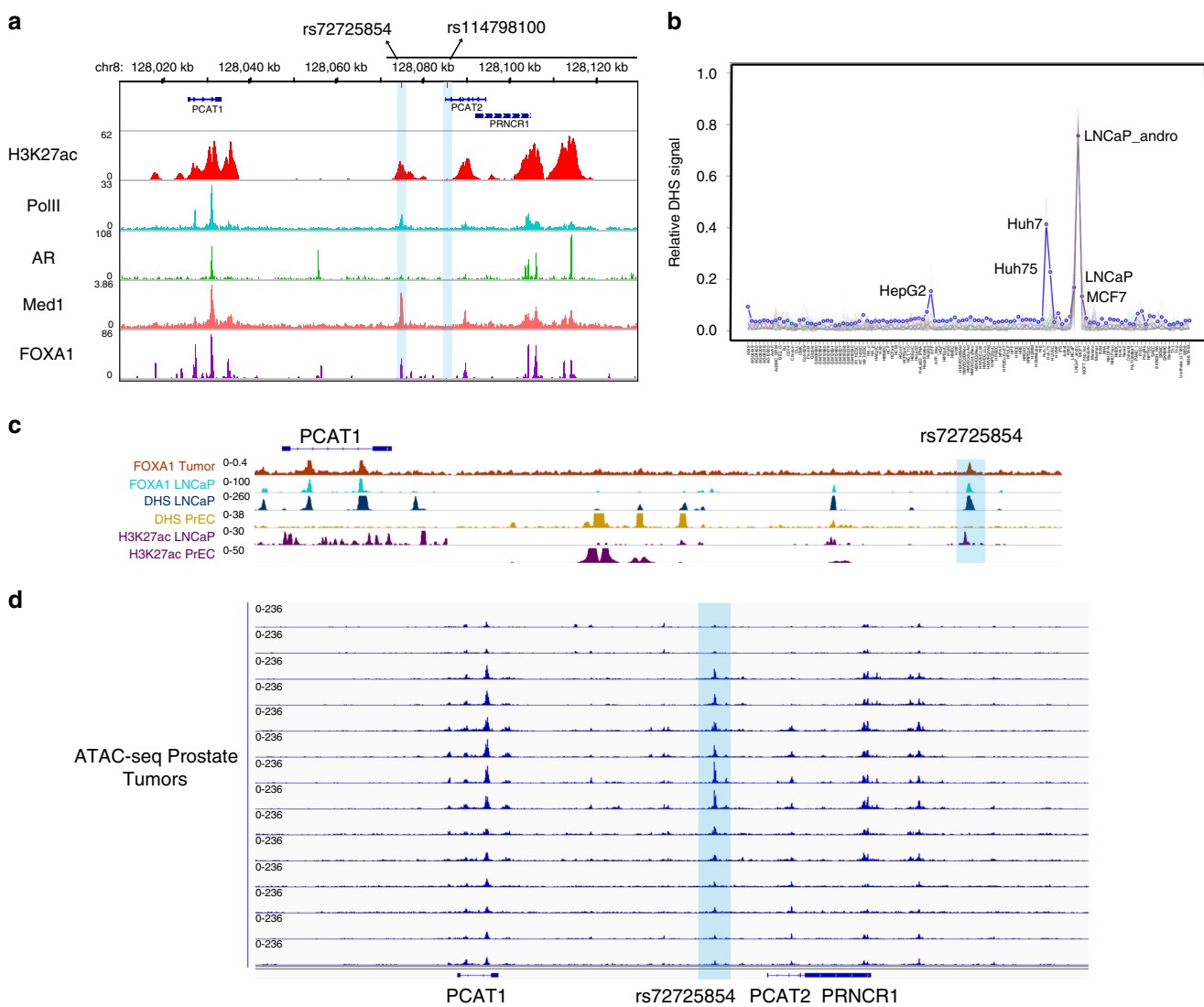

**Fig. 1 rs72725854 lies in a prostate cancer-specific enhancer. a** UCSC genome browser snapshot at rs72725854 region and its linked SNP rs114798100 showing ChIP-seq tracks for H3K27Ac, PolII, AR, Med1, and FOXA1. **b** Graph showing DHS signal around the rs72725854 region in various cell lines. **c** Genome browser snapshots show ChIP-seq signal of FOXA1 in prostate tumor, LNCaP cells; DHS and H3K27ac in LNCaP and prostate epithelial cells (PrEC). **d** IGV genome browser snapshot shows ATAC-seq signal around rs72725854 region in seven prostate adenocarcinoma tumor samples with two replicates each from TCGA.

Enhancers regulate target gene expression by chromatin looping, generally within the same topologically associating domain (TAD)[26,27]. To investigate whether the rs72725854-harboring enhancer physically interacts with any gene promoters in 8q24 region, we plotted the TAD structure around the rs72725854-harboring enhancer using publicly available data in LNCaP cells[28]. We observed a number of lncRNAs PCAT1, PCAT2, PRNCR1, CCAT1, CCAT2, CASC19, CASC21, PVT1, and the proto-oncogene MYC in the same TAD (Fig. 3a). However, only PCAT1, PRNCR1, CCAT1, PVT1, and MYC are actively transcribed in LNCaP cells as seen by H3K4me3 enrichment at the promoters of these genes though, multiple open chromatin signatures are present across the TAD. Next, to detect the looping targets of this enhancer, we performed circular chromatin-conformation capture (4C) assays at various anchors as the viewpoint. The 4C from the enhancer viewpoint exhibited significant interactions with the promoters of PCAT1 and PRNCR1 genes, located at ~41 kb upstream and ~18 kb downstream, respectively (Fig. 3b). To validate these interactions, we performed 4C at the PCAT1 promoter and found it to be interacting

with the rs72725854-harboring enhancer (Fig. 3c). In order to test whether the rs72725854-harboring enhancer forms long distance chromatin loops with MYC and PVT1 region at 730 kb distance, we performed the 4C at the PVT1 promoter (Fig. 3d). Not surprisingly, PVT1 promoter exhibited a short-range interaction with promoter of MYC but surprisingly, it also exhibited long-range interactions with PCAT1, PRNCR1, and rs72725854-harboring enhancer. However, we did not detect the significant interaction with CCAT1 promoter though it is transcribed in LNCaP cells (Fig. 3d). These data suggest that the rs72725854-harboring enhancer is in physical proximity with some but not all actively transcribing lncRNAs and coding gene MYC by several short- and long-range chromatin interactions. To check whether interactions in the TAD are conserved in other tumor cell lines that do not exhibit DHS at the rs72725854-harboring enhancer, we compared the HiC data from LNCaP with breast cancer cell line, T47D. Notably, the differential HiC matrices showed several new interactions in and around the 8q24 TAD in LNCaP compared to T47D (lower red pixels, Fig. 3e). Further, these interactions were between the PCAT1 (region 2) and PVT1/MYC

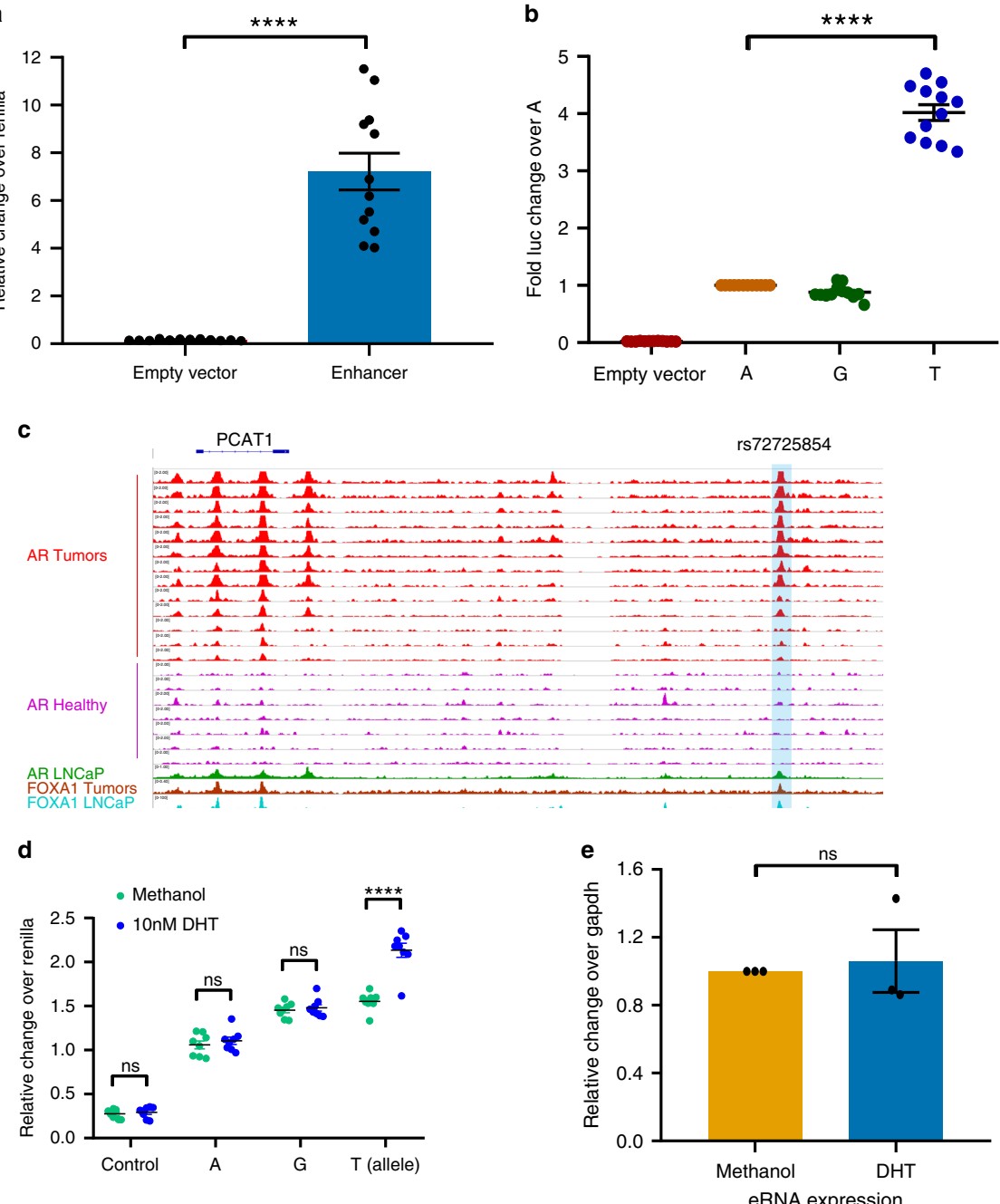

**Fig. 2 Risk allele sensitizes the enhancer to androgens. a** The graph depicts relative luciferase activity over Renilla for the empty plasmid and the plasmid harboring the rs72725854 enhancer region in LNCaP cells. **b** The graph depicts the fold change in reporter activity of empty plasmid, "G" and "T" allele over "A" allele in LNCaP cells. Error bars denote SEM from multiple biological replicates ($n > 3$) in **a** and **b**. **c** UCSC genome browser snapshot around the rs72725854 region with ChIP-seq tracks for AR in tumors, healthy individuals and LNCaP cells and, FOXA1 in tumors and LNCaP cells. **d** The graph depicts the relative change over Renilla for the empty plasmid and plasmids harboring the different alleles "A", "G", and "T" of rs72725854 enhancer in LNCaP cells grown in charcoal stripped media for three days including the treatment with methanol or 10 nM DHT for 16 h. Error bars denote SEM from multiple biological replicates ($n > 3$). **e** q-RT PCR analysis showing eRNA expression at rs72725854-harboring enhancer upon 10 nM DHT treatment for 16 h in LNCaP cells. Error bars denote SEM from three biological replicates. *p*-values were calculated by Student's two-tailed unpaired *t*-test in **a**, **b**, **d**, and **e** ****$p < 0.0001$, $^{ns}p > 0.05$. Source data are provided as a Source Data file.

(region 1) suggesting that 8q24 region exhibits extensive interactions in LNCaP where the enhancer is also active. Notably, the neighboring 5′ TAD that contains *FAM84B* gene, associated with prostatic neoplasms also exhibited prominent interactions with 8q24 TAD in LNCaP (Upper red pixels, Fig. 3e). Together, these chromatin-conformation data suggest that the rs72725854-harboring enhancer forms a spatial network with *PCAT1,*

*PRNCR1, PVT1,* and *MYC* genes in 3D nuclear space in prostate cancer cells.

After detecting the physical interactions among lncRNA genes and enhancer, we tested whether the enhancer regulates these genes transcriptionally. First, we interrogated the correlation between the DHS signal and PCAT1 expression in a number of cell lines from the ENCODE project and found that PCAT1

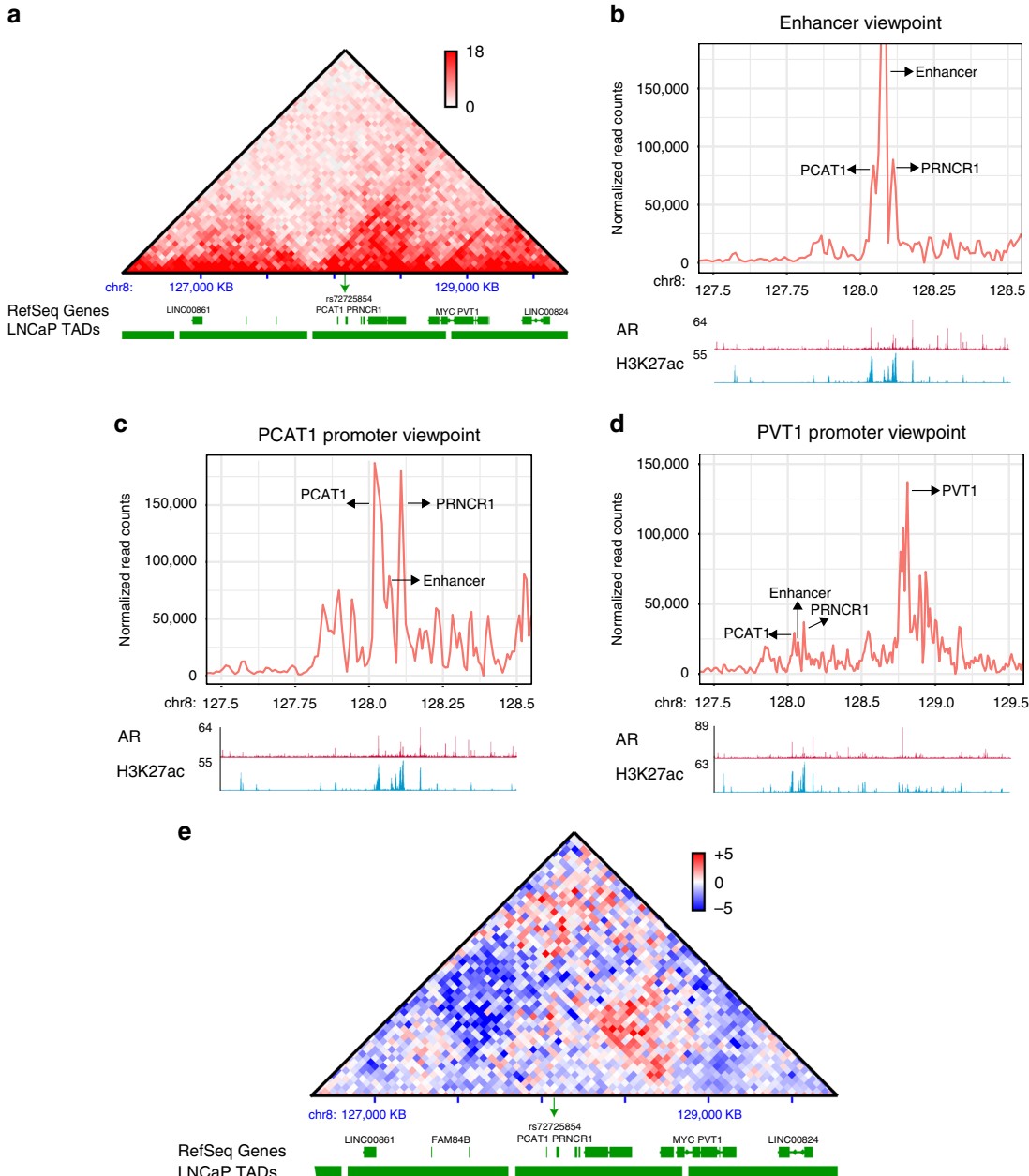

**Fig. 3 The enhancer physically interacts with lncRNA promoters. a** The TAD structure overlaid with gene annotations at the 8q24 region as seen from the HiC data in LNCaP cells. **b** Plots show 4C on the enhancer viewpoint using 4C-ker pipeline. **c** 4C plots from PCAT1 promoter viewpoint and **d** long-range interaction from 4C at PVT1 viewpoint, note the interactions marked by arrows. Tracks below the 4C plots in **b**–**d** show the ChIP-seq signal of AR and H3K27ac at the regions. **e** Comparative HiC in LNCaP and T47D showing the differential interactions at the 8q24 locus in the two cell lines. The red pixels show higher interactions in LNCaP and the blue ones show higher interactions in T47D, the heatmap is overlaid with gene annotations and TADs boundaries. 4C data are provided as source data file.

expression was highest in LNCaP compared to other cell lines where the DHS signal was almost negligible (Fig. 4a). Similarly, several studies have shown that the presence of eRNAs marks functional enhancers[24] and hence, we investigated whether there is a correlation between the eRNA expression from the enhancer and PCAT1 expression. We compared the nascent-RNA-sequencing derived eRNA expression from MCF7 and LNCaP cells. Unlike LNCaP, the DHS and eRNA expression is negligible in MCF7 cells (Supplementary Fig. 1b). The PCAT1 expression was also seen to be correspondingly low in MCF7 but was very high in LNCaP cells (Fig. 4b). These data suggest that PCAT1 is a probable transcriptional target of the enhancer.

To investigate whether enhancer suppression affects transcription of the non-coding genes implicated by the 4C data, we used CRISPRi to target this region. gRNAs on the SNP region were used in combination with dCas9-KRAB to block the enhancer in LNCaP cells (Supplementary Fig. 2a). We first confirmed the specificity of gRNAs targeting by testing the enrichment of dCas9 on the enhancer (Supplementary Fig. 2b). Further, we tested the functional efficacy of the enhancer block by evaluating the alterations in H3K9me3 and H3K27ac marks in the region. As expected, the H3K9me3 signal was significantly increased (Supplementary Fig. 2c), whereas the levels of H3K27ac were decreased (Supplementary Fig. 2d). As a result of these

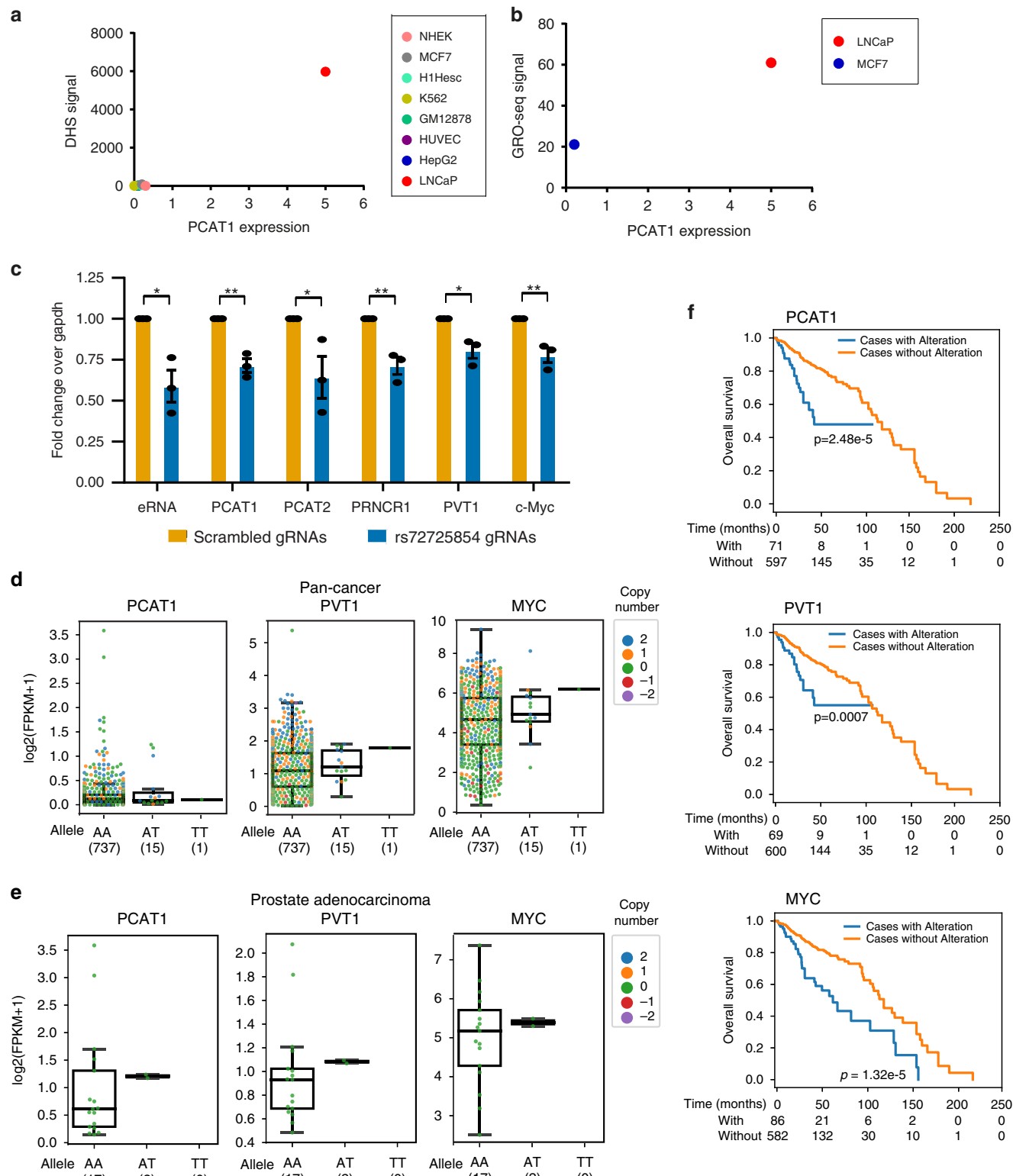

changes, the eRNA expression at the enhancer was significantly decreased in CRISPRi cells (Fig. 4c, first two bars) confirming the efficient blocking of the enhancer by gRNAs and dCas9-KRAB. Interestingly, the pre-mRNAs of PCAT1, PRNCR1, PVT1, and MYC were also downregulated upon enhancer blocking (Fig. 4c). However, the expression of other distal genes such as KLK2 and KLK3 were relatively unaffected (Supplementary Fig. 2e), demonstrating the specificity of the enhancer effects. These

results clearly suggest that rs72725854-harboring enhancer tran-scriptionally regulates these lncRNAs and MYC by virtue of 3D-chromatin architecture.

Next, we examined patient tumor samples to test the rela-tionship of risk allele "T" with the expression of the above-mentioned genes. Given that the rs72725854 variant is located in the non-coding region, we required patient samples with whole-genome sequencing/SNP-arrays (for genotype information) as

**Fig. 4 The enhancer transcriptionally regulates the lncRNA hub. a** Expression level of PCAT1 is plotted against the DHS intensity at rs72725854 regions across eight cell lines from ENCODE data. **b** Expression level of PCAT1 is plotted against the nascent-RNA signal derived from GRO-seq in LNCaP and MCF7 cell lines. **c** q-RT-PCRs show the relative fold changes of pre-mRNAs of various lncRNAs and MYC upon constitutive CRISPR blocking of the rs72725854 region with scr or specific gRNAs in LNCaP cells using dCas9-KRAB. Error bars denote SEM from three biological replicates. **d, e** Expression of PCAT1, PVT1, and MYC in patient tumor samples with different alleles of rs72725854 namely; AA, AT, and TT at pan-cancer level (from PCAWG) (**d**) and in prostate adenocarcinomas (from PCAWG) (**e**). Each dot represents a sample and the color indicates relative copy number status of the gene (0 is neutral, 1 is amplified, 2 is high-level amplified, −1 is deleted, −2 is deep deletion). The boxplots in **d** and **e** depict the minima (Q1-1.5*IQR), first quartile, median, third quartile, and maxima (Q3 + 1.5*IQR). **f** The survival analysis of prostate cancer patients exhibiting genetic alteration (copy number or mutations) vs. no alteration in PCAT1 (top), PVT1 (center) and MYC (bottom) genes irrespective of the genotype at rs72725854. The number of patients at each time interval in the cases with alteration (with) and the cases without alteration (without) of the respective gene is given below each plot. The data shown here were obtained from three different prostate cancer cohorts (see "Methods" and Supplementary Fig. 3f). p-values were calculated by Student's two-tailed unpaired t-test in **c**. *$p < 0.05$, **$p < 0.01$ and $^{ns}p > 0.05$. The p-values in **f** were calculated by log-rank test. Source data are provided as a Source Data file.

well as matched RNA-seq data to draw the association between rs72725854 alleles and target gene expression. Toward this, we explored all publicly available data sets (TCGA, ICGC, and PCAWG) and identified 753 patient samples across different tumor types from PCAWG. Of these, 737 samples had A/A genotype at the rs72725854 locus, whereas only 15 tumors were heterozygous (A/T) and one homozygous (T/T) for the risk allele. The small number of samples with "T" allele can be explained by the low frequency (~6%) of rs72725854 in African population, and that this population is not well represented in the available cancer cohorts. Interestingly, as compared to A/A genotype, tumors with A/T or T/T genotypes exhibited higher expression of PCAT1, PVT1, and MYC mRNAs (Fig. 4d). Further, within prostate adenocarcinomas of PCAWG cohort, 17 tumors had A/A genotype, only 2 were with A/T and none were with T/T genotype but, similar to the observation in the pan-cancer data, the expression of PCAT1, PVT1, and MYC was found to be higher in those with A/T genotypes (Fig. 4e). The effect of genotype on the expression of these genes was independent of their copy number status as most tumors having the A/T or T/T genotype did not exhibit aberrant copy number alterations (Fig. 4d, e, vertical panel with dots). Together, these data support the observation that the enhancer region harboring rs72725854 acts as an enhancer hotspot that not only regulates the neighboring genes such as PCAT1 but also the distant gene, PVT1. The MYC levels were also significantly higher in tumors with A/T and T/T genotypes which may be a direct effect or due to a positive regulatory loop between PVT1 and MYC transcription[29-31].

Further, we tested whether the alterations in these lncRNAs and MYC have any effect on the survival of the prostate cancer patients. Since the above sample set has low number of "T" allele, we evaluated all prostate cancer patients (available in cbioportal with exome/genome/SNP array data from multiple studies; see "Methods" section) who showed genetic alterations (copy number alterations or mutations) and no alterations in the PCAT1, PVT1 and MYC genes, irrespective of the ancestry and genotype at rs72725854. We found that most of the patients with alterations showed an amplification of the gene (Supplementary Fig. 2f), possibly contributing to overexpression of the lncRNAs and MYC. Independent of the "T" allele, the patients with these alterations showed significantly reduced 5 year and overall survival as opposed to the patients without any alterations (Fig. 4f) implicating, poor prognosis with the higher expression of these lncRNAs and MYC. In addition, the high PVT1 expressing prostate cancer patients also showed a higher Gleason score as compared to the subset of patients showing lower PVT1 expression (Supplementary Fig. 2g). Similarly, "T" allele of rs72725854 is associated with higher Gleason score of prostate tumors in African men[8]. Together, these results indicate that the enhancer region containing variant rs72725854 is involved in the

regulation of these lncRNAs and MYC and the risk allele "T" is correlated with the induced expression of PCAT1, PVT1, and MYC. Further, higher expression of these genes may lead to the poor survival in prostate cancer patients irrespective of their genotype at rs72725854 and ancestry.

Enhancers gain or lose their activity by virtue of binding with transcription factors. Thus, to biologically underpin the higher enhancer activity of the risk allele "T" (Fig. 2b), we tested the gain or loss of transcription factor (TF) motifs in the risk allele. The in silico TF motif search analysis using position weight matrix showed that the "T" allele created putative binding site for an ETS family transcription factor, SPDEF (Fig. 5a; Supplementary Fig. 3a). SPDEF is highly expressed in prostate cancer tissues and co-binds to the AR-bound sites on the DNA[32]; thus, transcriptionally activates the prostate-specific antigen gene (PSA)[33]. To confirm the in silico motif prediction, we performed electrophoretic mobility shift assay (EMSA) with active lysates from LNCaP cells and two sets of oligos differing only by the T vs. A base. Although, "A" oligos exhibited binding with an unknown protein in lysate, the "T" oligos showed a specific binding with a protein corresponding to the size of recombinant SPDEF protein (Fig. 5b). Whereas, the scramble oligo with same GC content did not exhibit such binding (Supplementary Fig. 3b). Further, the protein complex associated with "T" oligo was reduced upon knockdown of SPDEF with specific pool of siRNAs (Supplementary Fig. 3c), suggesting that the bound-complex on "T" oligo contains SPDEF. We investigated the binding preference of SPDEF on "T" allele further by first interrogating the binding of SPDEF on known binding regions in the genome such as PCAT1 intron[32], PSA promoter[33], and potential negative controls (NC1 and NC2) based on SPDEF ChIP-seq data[32]. Both, PCAT1 intron and PSA promoter showed the enrichment of SPDEF by several folds over the negative control regions (Fig. 5c). After validating the binding, we extended the binding strength analysis on independent plasmids carrying the enhancer region with the "A" or "T" alleles of rs72725854. The ChIP-qPCRs showed a preferential binding of SPDEF on the risk allele "T" with an enrichment comparable to PCAT1 intronic region, whereas the non-risk alleles "A" exhibited binding strength almost similar to negative controls (Fig. 5c). These data confirm the in silico motif analysis.

To test the transcriptional outcome of SPDEF binding to the "T" allele, we tested reporter activity upon the successful overexpression and knockdown of SPDEF (Fig. 5d). In reporter assays, activity of the "T" allele was down to baseline upon SPDEF knockdown, whereas the "A" allele was unaffected by such perturbations (Fig. 5e). Furthermore, upon overexpression of SPDEF, the enhancer activity of the "T" allele increased to ~80-fold (Fig. 5f), though the "A" allele also exhibited SPDEF binding upon overexpression likely due to the presence of weak SPDEF motif and SPDEF overexpression. These data suggest that the

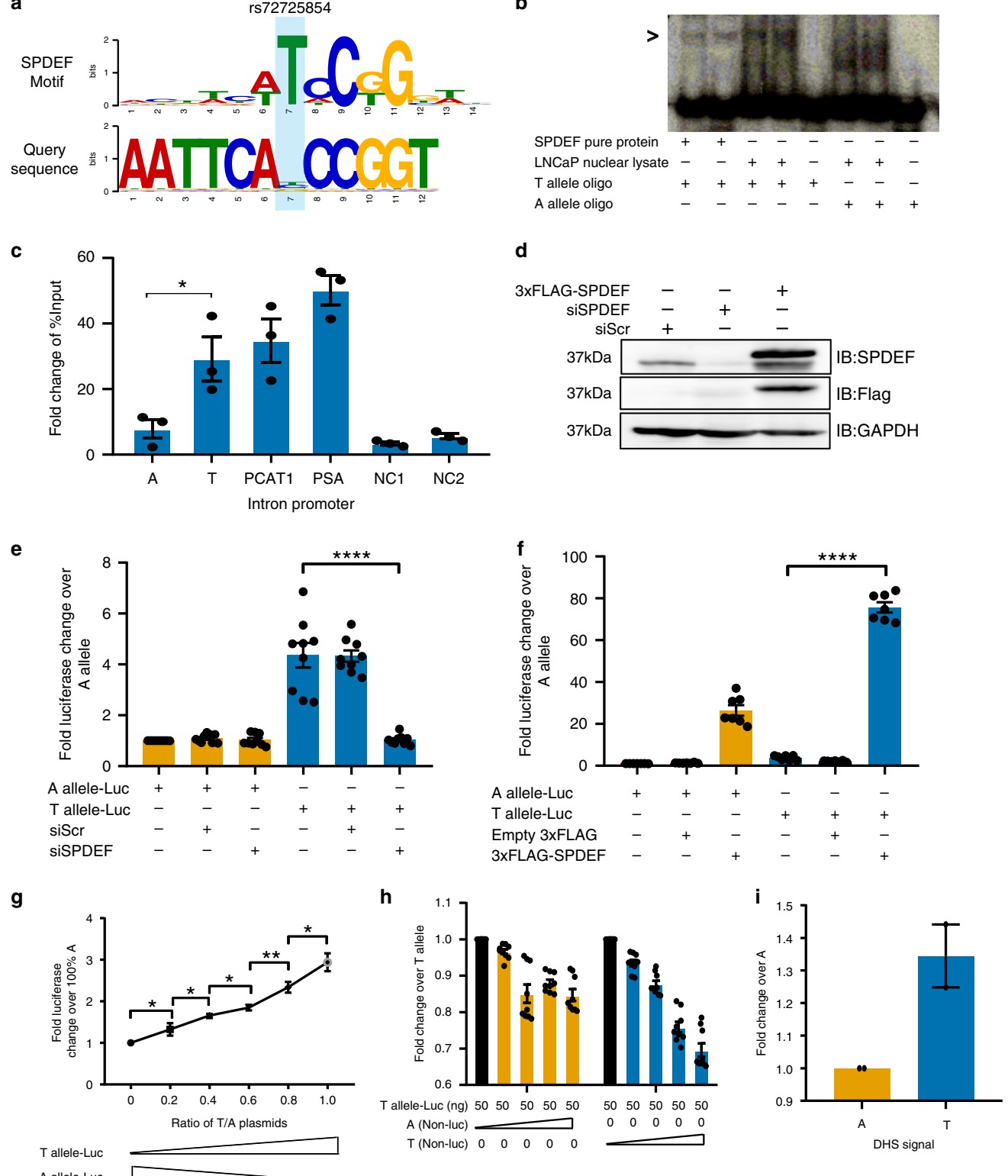

increased activity of the risk allele "T" was due to gain of SPDEF binding.

We also observed a dose-dependent increase in the reporter activity as the proportion of the "T" allele over "A" increased in the pool of both plasmids (Fig. 5g, method). This suggests that the higher reporter activity is being contributed by the "T" allele by virtue of more efficient recruitment of SPDEF. To determine whether the "T" allele can compete better for SPDEF binding as

compared to the "A", LNCaP cells were co-transfected with definite amounts of "T" allele plasmid that has luciferase reporter gene ("T" allele-luc) and increasing doses of plasmid with either "A" or "T" alleles but without luciferase. If SPDEF binds more efficiently on "T" as compared to "A" allele, then it will compete more effectively with the SPDEF complex bound on "T" allele-luc resulting in loss of its luciferase activity. Indeed, the "T" allele-luc exhibited linear loss of activity as the doses of "T" allele without

**Fig. 5 Risk allele increases prostate cancer risk by gain of SPDEF. a** Position weight matrix analysis shows a gain of strong motif of SPDEF in the risk allele "T" of rs72725854. **b** EMSA showing differential affinities of the risk "T" and non-risk "A" alleles of rs72725854 for SPDEF. The experiment was performed thrice. **c** The graph depicting FLAG-SPDEF ChIP-qPCRs on plasmids harboring different alleles of rs72725854, PCAT1 intron, PVT1 promoter, and two negative controls in LNCaP cells overexpressing 3xFLAG-SPDEF. **d** Immunoblot with anti-SPDEF, FLAG, and GAPDH antibodies showing the levels of SPDEF and GAPDH proteins upon overexpression and the knockdown of the SPDEF protein by 3xFLAG-SPDEF and on-target siRNA pools, respectively. The experiment was performed thrice. **e** Reporter assays show the alterations in activities of non- risk and risk alleles upon specific knockdown of SPDEF. **f** Reporter assays show the alterations in reporter activities of non-risk and risk alleles upon SPDEF overexpression. **g** Reporter assays show the change in the luciferase activity in a dose-dependent manner when the percentage of the plasmid with the "T" allele increases over "A" allele in the pool of "A" and "T" alleles. **h** Competition reporter assays show the alterations in reporter activity of "T allele-Luc" upon dose-dependent (0, 50, 150, 250, 350 ng) overexpression of "A (Non-Luc)" or "T(Non-Luc)" plasmids. **i** Graph depicting qPCR signals on the "A" and "T" plasmids in DNase I hypersensitivity assays performed in LNCaP cells ($n = 2$). Error bars denote SEM from three biological replicates in **c**, $n > 3$ replicates in **e**–**h**. *p*-values were calculated by Student's two-tailed unpaired *t*-test in **c**, **e**, **f** and **g**. *$p < 0.05$, **$p < 0.01$, ***$p < 0.001$, ****$p < 0.0001$, [ns]$p > 0.05$. Source data are provided as a Source Data file.

luciferase increased (Fig. 5h, blue bars). "A" allele also showed competition however, it did not follow a pattern of linear loss even with the highest doses of "A" (Fig. 5h, orange bars) suggesting, random binding events of SPDEF on "A" allele due to its overexpression. Similarly, the "T" allele exhibited more open chromatin features as compared to "A" allele as seen by DNase I hypersensitivity assay (Fig. 5i). Together, these results (Fig. 5c–i) suggest that the risk allele "T" leads to a stronger enhancer activity due to the gain of stronger SPDEF binding.

SPDEF binding to the risk allele activates the rs72725854-harboring enhancer, suggesting that SPDEF recruits activating machinery to the risk allele. Further, the risk allele exhibits a response to DHT indicating that SPDEF renders the risk allele sensitive to androgens. DHT activates AR recruitment on chromatin and SPDEF has been suggested to collaborate with AR[32], so we contrasted AR binding at SPDEF-bound and -unbound sites genome-wide, using available AR and SPDEF ChIP-seq data. Interestingly, a higher enrichment of AR was found on the SPDEF-bound sites over an equal number of random non-SPDEF AR sites, all non-SPDEF AR sites, random AR sites and all AR sites (Fig. 6a). To determine whether SPDEF and AR co-occurring peaks have a functional relevance, we investigated the presence of H3K27ac marks and PolII occupancy at these peaks vs. all AR peaks. We observed that SPDEF-AR co-occurring sites have higher AR, H3K27ac, and PolII signal than all AR peaks (Fig. 6b–d). The data suggests that AR binding in the genome is robust at SPDEF-bound sites and these sites are relatively more active than other AR sites as seen by high PolII. Gene ontology (GO) term analysis on genes near these regions exhibited morphogenesis, prostate bud, mammary gland development and mesenchymal cell proliferation as key terms (Fig. 6e). Whereas, AR alone regions showed metabolic processes as a predominant category. To confirm that AR is recruited to the "T" allele via SPDEF, we tested the enhancer activity of the "T" allele upon DHT treatment in the presence and absence of SPDEF by siRNA knockdown. As expected, the DHT response was completely abolished in the absence of SPDEF (Fig. 6f), suggesting that the DHT response of the "T" allele is due to the gain of SPDEF binding. Expression of SPDEF is highly induced in prostate and breast cancer origin cell lines as opposed to other cancer types, suggesting the specificity of SPDEF to prostate cancer (Fig. 6g). The enhanced expression of SPDEF could be due to its genetic alterations in prostate tumors (Fig. 6g). Also, SPDEF is over-expressed in Prostate adenocarcinomas from TCGA cohort as compared to normal tissues both from TCGA and GTEx, supporting the role of SPDEF in prostate cancer progression and development (Fig. 6h). Further, ERG overexpression (ETS factor) in RWPE cells enhances the TAD structure at 8q24 locus by favouring the intra-TAD interactions (Supplementary Fig. 4), suggesting that the gain of ETS factor binding at regulatory regions has a potential to alter the three-dimensional chromatin

architecture[34]. Taken together, these data suggest that SPDEF bound to the risk allele influences sensitivity to DHT, resulting in greater enhancer activity and the regulation of nearby lncRNAs and MYC due to pre-established 3D genomic proximity (Fig. 6i).

## Discussion
Cancer is manifested by several oncogenes that gain enhancers upstream or downstream to their promoters[35–38]. In most cases, these enhancers control single neighboring gene but in some cases they regulate multiple genes over short and very long distances by virtue of 3D-chromatin folding[39]. Such mechanisms of enhancer function are poorly understood. Further, how rare genetic variation affects these enhancers is unknown. Here we identify one such enhancer in a non-coding region of the 8q24 locus, which is inactive in normal prostate tissues. However, the region gains enhancer activity in prostate tumors and in the prostate cancer cell line, LNCaP as seen by the presence of DHS, H3K27ac, FOXA1, and AR occupancies (Figs. 1 and 2). The enhancer region has regulatory potential observed in reporter assays with transcriptional strength increased when the risk allele "T" is present. Interestingly, the enhancer strength is enhanced by DHT treatment in case of risk allele "T", whereas the non-risk allele "A" is not responsive to androgens. This suggests that the risk allele might accelerate prostate cancer development in the risk allele carrying individuals as testosterone levels begin to rise following puberty.

Interestingly, unlike most enhancers that regulate only a single gene, the rs72725854-harboring enhancer is observed to regulate the transcription of multiple disease-associated lncRNAs and proto-oncogene MYC in the 8q24 region via 3D-conformation (Fig. 3). This enhancer forms chromatin loops with the genes within the same TAD thus forming a enhancer-promoter hub. These interactions within the hub result in the regulation of lncRNAs and MYC by rs72725854-harboring enhancer as observed by CRISPR studies (Fig. 4c). The upregulation of PCAT1, PRNCR1, PVT1, and MYC potentially confers the risk of cancer as their upregulation is strongly associated with prostate cancer[10,12,40–42].

We demonstrate that strong enhancer activity is due to the gain of a SPDEF motif created by the risk allele "T". The motif analysis at the enhancer region did not show a cognate motif for AR and the "T" allele exhibits no response to DHT upon SPDEF knockdown (Fig. 6f). Thus, the indirect recruitment of AR via SPDEF by the "T" allele seems to be the mechanism by which the enhancer elicits a DHT response. SPDEF has been shown to act like a transcriptional activator by directly binding to the DNA, and upon androgen exposure it recruits AR in trans via its DNA-binding domain. Thus, SPDEF enhances androgen-mediated activation of target genes[33]. Similarly, upregulation of SPDEF expression was seen to be associated with poor prognosis in prostate cancer[43]. These data suggest that upon ligand addition,

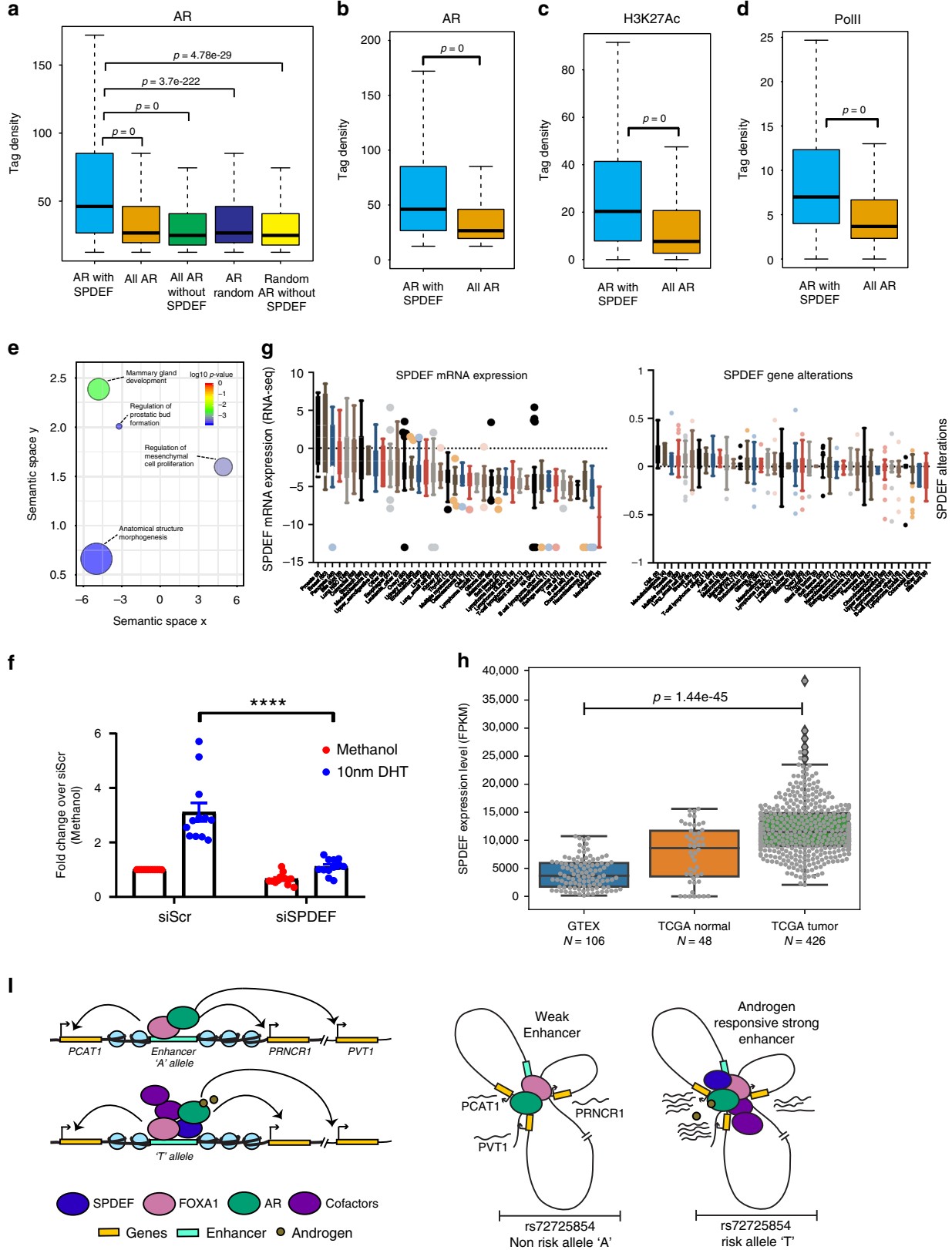

AR recruits activators that further intensify the transcriptional potential of both type of enhancers namely, bound by AR directly or in trans. We also found that the expression of lncRNAs and MYC is higher in tumors with the risk allele "T" (Fig. 4d, e) and that the risk allele enhances its transcriptional activity by responding to androgens.

In summary, we implicate a mechanism by which the most significant prostate cancer susceptibility variant for men of African ancestry alters risk. The integrative approach described in this study can be further used to assign functions to more non-coding variants in future studies, which is a primary task in the post-GWAS period.

**Fig. 6 SPDEF collaborates with AR to induce prostate-specific genes. a** Tag density plots show the relative levels of AR on various combinations of AR and SPDEF peaks mentioned on X axis. Tag density plots suggest a higher density of AR (**b**) H3K27ac (**c**), and PolII (**d**) occupancy on the AR-SPDEF co-occurring peaks as compared to all AR peaks. The boxplots in **a**–**d** depict the minima (Q1-1.5*IQR), first quartile, median, third quartile and maxima (Q3 + 1.5*IQR). p-values were calculated by Wilcoxon two-sided rank sum test in **a**–**d**. **e** GO term analysis on genes nearby SPDEF-AR co-occurring peaks indicates that these regions are involved in the development programs including prostate. Position of GO terms on the X and Y axis represents the relative closeness between GO terms based on GO graph structure. Size of circles represents the frequency of a particular GO term in the GO database. Color of the circle represents multiple test corrected p-value, see legend on the top right. **f** Reporter assays showing the change in DHT response of the "T" allele upon SPDEF knockdown. Error bars denote SEM from n > 3 biological replicates. p-value was calculated by Student's two-tailed unpaired t-test (****p < 0.0001). **g** SPDEF RNA expression and alterations across different cell lines from CCLE data sets. **h** SPDEF RNA expression levels plotted across different data sets, prostate normal tissue from GTEx, prostate adenocarcinomas and adjacent normal tissues from TCGA. p-value was calculated using Wilcoxon two-sided rank sum test. **i** Model depicts that the rs72725854 region functions like an enhancer even in the non-risk allele. The enhancer exhibits closer proximity to the lncRNA genes PCAT1, PRNCR1, and PVT1. Enhancer with risk allele recruits SPDEF, which in turn brings AR-activating machinery in response to DHT, resulting in the hyper-activation of the enhancer. Full enhancer activation robustly induces expression of target lncRNAs and MYC promoters that are already in 3D proximity. lncRNA and MYC activation potentially predisposes these individuals to prostate cancer. Source data are provided as a Source Data file.

## Methods

**Cell culture and treatment**. LNCaP, 293FT, and MCF7 cell lines were obtained from the American Type Culture Collection (ATCC). The cells were cultured in media and conditions as recommended by ATCC at 37 °C and 5% $CO_2$ in a humidified incubator. No mycoplasma contamination was detected in these cell lines by MycoAlert, Mycoplasma Detection kit (LT07-118, Lonza). For DHT response, LNCaP cells were serum starved for 72 h in Stripping medium (RPMI without phenol red with 10% charcoal stripped FBS and 1% Pen–Strep) including 16 h of stimulation with DHT (10 nM) (Sigma Inc.).

**Antibodies**. SPDEF (sc-166846, Santacruz Biotechnology), FLAG (F7425, Sigma), H3K27ac (ab4729, Abcam), H3K9me3 (39161, Active Motif), and GAPDH (sc-32233, Santacruz Biotechnology) antibodies were obtained from respective manufacturers.

**siRNA transfection**. siRNAs SMARTpools specifically targeting SPDEF (L-020199-00-005) and scrambled siRNA were purchased from GE Dharmacon. Lipofectamine 2000 (Invitrogen) was used for siRNA transfections as per manufacturer's recommendations.

**Dual-luciferase reporter assays and site-directed mutagenesis**. A 251-bp fragment with 125 bp on either side of the SNP was cloned as a core enhancer region which was also marked by H3K27ac and FOXA1 occupancy in LNCaP. The region was PCR amplified from LNCaP cells that have the "A" allele. The "G" and the "T" alleles of the SNP were created using single base changes in internal forward primers at the SNP region and the region was amplified with reverse oligos. These mega-primers were used with the forward primers, the amplicons were gel purified and cloned in pGL4.23 reporter plasmid using KpnI and XhoI sites. Constructs were co-transfected with pRL-TK vector containing Renilla luciferase into LNCaP cells using Lipofectamine 2000 (Invitrogen). The cells were cultured in normal RPMI media or in stripping media for 72 h including DHT treatment (10 nM) for 16 h to check the DHT response. The cells were harvested 36 h post transfection in all of the assays. Luciferase activity was estimated using the Dual-Luciferase Reporter Assay System (Promega) with Renilla luciferase (Rluc) as the internal reference. Multiple biological replicates but no technical replicate was performed. The p-values were calculated by Student's two-tailed unpaired t-test.

**Gradient reporter assay**. The "A" and "T" allele plasmids were transfected into LNCaP cells in a 24-well plate in different proportions keeping the total plasmid amount the same (300 ng). The "T" allele plasmid was added in increasing amounts while "A" allele was decreased in following way—0 ng T + 300 ng A, 60 ng T + 240 ng A, 120 ng T + 180 ng A, 180 ng T + 120 ng A, 240 ng T + 60 ng A, and 300 ng T + 0 ng A, such that the proportion of T increases from 0, 20, 40, 60, 80, and 100% and the proportions of A allele decreases in same manner. Luciferase activity was estimated using the Dual-Luciferase Reporter Assay System (Promega) with Renilla luciferase (Rluc) as the internal reference for data normalization. All the values were normalized to the samples where the proportion of A:T was 100:0 (300 ng A plasmid and 0 ng T plasmid). The p-values was calculated by Student's two-tailed unpaired t-test.

**Competition reporter assay**. The luciferase gene was excised out from the A and T plasmids in pGL4.23 and also from the empty pGL4.23 plasmid. Each well in a 24-well plate was transfected with a background T allele plasmid with luciferase (T allele-luc) (50 ng). Increasing amounts of plasmids (A or T) without luciferase were added, 0, 50, 150, 250, and 350 ng to compete with this background plasmid. The empty pGL4.23 plasmid without luciferase was used to maintain the overall

plasmid amount to 450 ng. 3 ng/well Renilla-TK plasmid was transfected for normalization. Luciferase activity was estimated using the Dual-Luciferase Reporter Assay System (Promega) with Renilla luciferase (Rluc) as the internal reference. All the values were normalized to the sample with 50 ng T allele with luciferase. Multiple biological replicates but no technical replicate was performed. The p-values was calculated by Student's two-tailed unpaired t-test.

**Chromatin immunoprecipitation (ChIP)**. ChIP assays were performed using LNCaP cells treated with 10 nM DHT for 16 h in 10 $cm^2$ dishes at ~80% confluency. Cells were crosslinked by 1% formaldehyde for 10 min, and the reaction was quenched with 125 mM glycine for 5 min. After washing with cold PBS (3×), the cells were scraped and pelleted at 2500×g for 10 min. The pellet was resuspended in nuclear lysis buffer (Tris-HCl pH 7.4 (50 mM), SDS (1%), EDTA pH 8.0 (10 mM), PIC (1×)). The solution was subjected to sonication using Diagenode Bioruptor Pico for 20 cycles (30 s ON and 30 s OFF) to generate fragments of around ~500 bp. The solution was cleared by centrifugation at 12,000×g for 12 min at 4 °C. The concentration of the supernatant was measured and 100 μg chromatin was used for each IP. The lysates were diluted using 1.5 fold dilution buffer (Tris-HCl pH 7.4 (20 mM), NaCl (100 mM), EDTA pH 8.0 (2 mM), Triton X-100 (0.5%), PIC (1×)), and antibodies (1 μg) were added to the IP samples and rotated overnight at 4 °C. Pre-blocked (with 1% BSA) Protein G dynabeads (10004D, Invitrogen) were added to the tubes and rotated for 4 h at 4 °C. The samples were washed first with Wash buffer I [Tris-Cl pH 7.4 (20 mM), NaCl (150 mM), SDS (0.1%), EDTA pH 8.0 (2 mM), Triton X-100 (1%)] by rotation for 5 min at 4 °C, followed by Wash buffer II [Tris-Cl pH 7.4 (20 mM), NaCl (500 mM), EDTA pH 8.0 (2 mM), Triton X-100 (1%)], followed by Wash buffer III [Tris-Cl pH 7.4 (10 mM), LiCl (250 mM), NP-40 (1%), sodium deoxycholate (1%), EDTA pH 8.0 (1 mM)] and lastly by TE (pH 8.0). The chromatin was then eluted in Elution buffer [sodium bicarbonate (100 mM) and SDS (1%)] for 30 min at 37 °C in a thermomixer. The supernatant was collected in separate tubes and 14 μl NaCl (5 M) was added to 200 μl of eluted samples and kept at 65 °C overnight for de-crosslinking. The samples were subjected to phenol:chloroform:isoamylalcohol (PCI) purification followed by ethanol precipitation and each sample was eluted in TE (pH 8.0). These samples were then used for quantitative RT-PCRs. In the ChIP-qPCRs on plasmids, 3xFLAG-SPDEF was overexpressed in LNCaP and the IP was performed by the FLAG antibody (F7425).

**RNA isolation, cDNA synthesis, and quantitative PCRs**. RNA was isolated using Trizol (Invitrogen) as per manufacturer's recommendation. 2 μg of RNA was used for each cDNA synthesis using random hexamers by Superscript IV (Invitrogen) as per manufacturer's recommendation. ABI SYBR green was used and the protocol and program used were as per manufacturers' recommendations. The CFX96 touch (Biorad) real time PCR was used for qPCR runs. qPCRs were performed using three biological replicates for each sample. The fold changes were calculated by ΔΔCt method as described in https://www.thermofisher.com/in/en/home/life-science/epigenetics-noncoding-rna-research/chromatin-remodeling/chromatin-immunoprecipitation-chip/chip-analysis.html. The p-values were calculated by Student's unpaired two-tailed t-test.

**Circular chromosome conformation capture (4C)**. 4C was performed as per the protocol described in van de Werken et al.[44], with minor variations. LNCaP cells were fixed with fresh formaldehyde (1.5%) and quenched with glycine (125 mM) followed by washes with ice-cold PBS (2×) and scraped, pelleted and stored at −80 °C. Lysis buffer [Tris-Cl pH 8.0 (10 mM), NaCl (10 mM), NP-40 (0.2%), PIC (1×)] was added to the pellets and were homogenized by Dounce homogenizer (15 stroked with pestle A followed by pestle B). The 3C digestion was performed with HindIII (400 units, NEB) and ligation was performed by the T4 DNA ligase and 7.61 ml ligation mix (745 μl 10% Triton X-100, 745 μl 10x ligation buffer

(500 mM Tris-HCl pH7.5, 100 mM MgCl$_2$, 100 mM DTT), 80 µl 10 mg/ml BSA, 80 µl 100 mM ATP and 5.96 ml water). The ligated samples were de-crosslinked overnight then purified by PCI purification and subjected to ethanol precipitation and the pellet was eluted in TE (pH 8.0) to obtain the 3C library. The second 4C digestion was performed by *Dpn*II (50 units, NEB) and the samples were ligated, purified and precipitated similar to the 3C library to obtain the 4C library. The 4C library was subjected to RNAse treatment and purified by the QIAquick PCR purification kit. The concentration of the library was then measured by Nanodrop and subjected to PCRs using the oligos for the respective viewpoints. The oligos used for the SNP, PCAT1 and PVT1 viewpoint are mentioned in the (Supplementary Table 1). The samples were PCR purified and subjected to next-generation sequencing with Illumina HiSeq2500 using 50 bp single-end reads (Supplementary Table 3). Data analysis was performed using 4C-ker (https://github.com/rr1859/R.4Cker) using default parameters[45].

**CRISPR/Cas9 guide selection and cloning**. gRNAs were designed using CRISPR Design tool (crispr.mit.edu). The gRNAs with the best score were chosen with least off-targets. We used pgRNA-humanized vector (was a gift from Stanley Qi (addgene #44248)) for gRNA cloning using *Bst*XI and *Xho*I restriction sites.

**CRISPRi blocking**. 293FT cells were co-transfected with VSVG, a gift from Bob Weinberg (addgene #8454) and Pax2 plasmid, was a gift from Didier Trono (addgene #12260), along with the gRNAs cloned in pgRNA-humanized plasmid (Supplementary Table 1) and Lenti dCas9-KRAB, was a gift from Kristen Brennand (addgene #99372) with lipofectamine 2000. Lenti-viruses collected in two rounds, 48 and 72 h post transfection with media change after the first collection. The pooled viruses were then filtered with a 0.44 µm syringe filter and added along with polybrene (8 µg/ml) to LNCaP cells. The transduction was carried out for 24 h and puromycin selection (3 µg/ml) was started after it. The selection was carried out till 5 days until all cells in the control plate died. The cell lines were grown for a few generations and later used for experiments. ChIP-qPCRs were performed for Flag-dCas9, H3K27ac, and H3K9me3 marks to ensure enhancer blocking. RNA isolation and cDNA preparation was carried out in these cell lines as mentioned above and qPCRs for ChIP and RT-PCRs were performed (Supplementary Table 1).

**SPDEF cloning and protein purification**. SPDEF cDNA was cloned in pGEX6p1vector (Addgene #27-4591-01) using *Eco*RI and *Bam*HI restriction sites. The oligos used in the cloning are mentioned in (Supplementary Table 1). The recombinant protein was expressed in Rosetta strain. The eluted protein was subjected to buffer exchange using protein concentration columns. The GST tag was cleaved before using for EMSAs. For the overexpression of SPDEF in LNCaP cells, SPDEF cDNA was cloned downstream of the 3xFLAG tag in the pcDNA 3xFLAG CMV10 vector using BamHI and EcoRI restriction sites (Supplementary Table 1).

**Electrophoretic mobility shift assay**. The nuclear lysate used in EMSAs was obtained from LNCaP cells treated with 10 nM DHT for 16 h. LNCaP cells were harvested in ice-cold PBS and nuclei were isolated in Nuclei isolation buffer [Tris-Cl pH 7.5 (40 mM), MgCl$_2$ (20 mM), Sucrose (1.2 M), Triton X-100 (4%), PIC (1×)]. Further, nuclear lysate was made in RIP buffer [Tris-Cl pH 7.4 (25 mM), KCl (150 mM), NP-40 (0.5%), PIC (1×)] and sonicated for 10 cycles (30 s ON and 30 s OFF) using bioruptor. The lysate was then cleared by centrifugation and the cleared supernatant was used for EMSAs.

51 nucleotides long oligos with A or T at 26th position flanking rs72725854 and its reverse complementary oligos (Supplementary Table 1).were annealed by heating to 95 °C and slow cooling to room temperature. The annealed oligos were purified from native PAGE gel before end labeling with γP$^{32}$ATP. The labeled oligos were then incubated with LNCaP nuclear lysate, or with scr or siSPDEF lysates, or with pure SPDEF protein and were run on a 6% native polyacrylamide gel, the gel was exposed and was then imaged.

**Cell line data acquisition**. The source of ChIP-seq data for SPDEF, H3K27ac, H3K4me1, PolII, AR, FOXA1, GRO-seq, and DHS-seq data in LNCaP, VCAP, and MCF7 are mentioned in (Supplementary Table 2). The RNA-seq and DHS-seq for different cell lines was obtained from ENCODE consortium.

**ChIP-seq data analysis**. The sequenced reads were aligned to hg19 assembly using default Bowtie2 options. Tag directories were made from the aligned reads to identify ChIP-seq peaks using HOMER[46]. A 200 bp sliding window was used to identify narrow peaks, which are characteristic of transcription factor peaks. The common artifacts from clonal amplification were neglected as only one tag from each unique genomic position was considered. The threshold was set at a false discovery rate (FDR) of 0.001 determined by peak finding using randomized tag positions in a genome with an effective size of $2 \times 10^9$ bp. For ChIP-seq of histone marks, seed regions were initially found using a peak size of 500 bp (FDR < 0.001) to identify enriched loci. Enriched regions separated by <1 kb were merged and considered as blocks of variable lengths. The read densities as bed graph files were

calculated across the genome and this track was uploaded to UCSC genome browser. HOMER annotatePeaks.pl was used to quantify the normalized tag counts of different data sets from specific regions.

**Motif analysis**. Motif analysis and *p*-values were obtained using Tomtom[47] from the meme suite to test the gain or loss of transcription factor motifs when different alleles of rs72725854 are present.

**DNase I hypersensitivity assay-qPCR**. Cells were Transfected with A or T plasmid using lipofectamine 2000. Post 6 h of transfections, cells where treated with DHT for 16 h. Cells were trypsinized, quenched with media, and pelleted down at 200×g for 5 min at 4 °C. Pellet was then washed twice with ice-cold PBS and were then resuspended in 250 µl ice-cold DNase1 buffer (HEPES pH.8 10 mM, KCl 50 mM, MgCl$_2$ 5 mM, CaCl$_2$ 3 mM, NP-40 1%, Glycerol 8%, DTT 1 mM). Cells were Dounce homogenized (10 strokes) to isolate nuclei. 80 or 160 U/ml of DNase1 (Roche 04716728001) was added and incubated at 25 °C water bath for 3 min. Immediately, 300 µl of DNase1 Stop buffer (EGTA 20 mM, SDS 1%) was added to terminate the reaction. RNase A (10 mg/ml) was added to each sample for 2 h at 37 °C, and then with Proteinase K (10 mg/ml) at 55 °C overnight. DNA was purified using Ph:Chl:IAA and was size selected using AMPURE XP beads. The Ct for ampicillin gene (Supplementary Table 1) was used to normalize the Ct from rs72725854 region.

**Comparative HiC**. The HiC data sets were analyzed using the Juicer pipeline[48]. Hi-C reads were aligned to the hg19 human reference genome with the appropriate restriction site. The.hic file generated from the juicer pipeline was then visualized using Juicebox. The contact maps were generated at a 50 kb resolution using balanced normalization (Knight–Ruiz balancing algorithm). The differential contact maps were generated using the observed/control option.

**Gene expression analysis in tumor samples**. Genotype information at rs72725854 locus, gene expression and copy number data from the whole-genome sequencing and RNA-sequencing of pan-cancer ($n = 753$) and prostate adenocarcinomas ($n = 19$) (shown in Fig. 4d, e) were obtained from PCAWG (https://dcc.icgc.org/pcawg[49]).

**Survival analysis**. Survival plots were made using lifelines package[50] in python (https://lifelines.readthedocs.io/en/latest/index.html). The plots (Fig. 4f) comparing altered (samples with mutation and copy number changes) versus non-altered were made using combined data from three cohorts: prostate adenocarcinoma (TCGA, PanCancer Atlas) ($n = 494$), metastatic prostate adenocarcinoma (SU2C/PCF Dream Team, PNAS 2019) ($n = 444$), and metastatic prostate adenocarcinoma (MCTP, Nature 2012) ($n = 61$) from cbioportal (https://www.cbioportal.org/).

**ATAC-seq data in prostate tumors**. ATAC-seq data in prostate adenocarcinoma tumors (Fig. 1d) were obtained from TCGA study.

**SPDEF expression in normal and tumor tissue**. Normalized RNA-seq data of normal and tumor tissue was obtained from Wang et al.[51]. The RNA expression (FPKM) of SPDEF in prostate normal (GTEx), TCGA prostate adenocarcinoma, and adjacent normal are shown in Fig. 6h.

**GO term analysis**. gprofiler[52] was used to obtain enriched GO terms for biological processes with *p*-value 0.005 (multiple test corrected, g:SCS). REVIGO[53] was used to visualize the enriched GO terms.

**Reporting summary**. Further information on research design is available in the Nature Research Reporting Summary linked to this article.

## Data availability
Mutations, gene expression, and clinical annotation of cancer patient samples were obtained from the publicly available repositories: PCAWG (https://dcc.icgc.org/pcawg), TCGA Genomic Data Commons (GDC) (https://portal.gdc.cancer.gov/), and cBioPortal (https://www.cbioportal.org/). The ATAC-seq data of prostate tumors in TCGA were obtained from https://gdc.cancer.gov/about-data/publications/ATACseq-AWG. Normalized RNA-seq data of normal and tumor tissues (in TCGA and GTEx) were obtained from Wang et al.[51]. The cancer cell line data were obtained from CCLE (https://portals.broadinstitute.org/ccle) and ENCODE (https://www.encodeproject.org/). A reporting summary for this article is available as a supplementary information file. Source data are provided with this paper.

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

## Acknowledgements

We thank Tapas K. Kundu, Aswin Sai Narain Seshasayee, and DN lab members for discussions. We thank Amanjot Singh for critical inputs in study design. We thank Sakshi Gorey for preparing recombinant SPDEF proteins. We acknowledge support of the Department of Atomic Energy, Government of India, under project no. 12-R&D-TFR-5.04-0800 and intramural funds from NCBS-TIFR (to D.N., P.V.S., and R.S.). D.N. is EMBO Global Investigator. We also acknowledge funding support from Welcome-IA: IA/1/14/2/501539 and SERB:CRG/2019/005714 (to D.N.); R01 CA165862 and U19CA214253 (to C.A.H.); Ramanujan Fellowship (SERB, SB/S2/RJN-071/2018) (to R.S.); DOD PC180367, NIH R01CA193910 and NIH R01CA227237 (to M.L.F.). K.W. is supported by Shyama Prasad Mukherjee fellowship from CSIR, India. U.F. is supported by Dept. of Biotechnology (Govt. of India) junior research fellowship. A.K.S., D.S., and R.M. are supported by NCBS/TIFR graduate program.

## Author contributions

Study was conceived and designed by K.W. and D.N. K.W. performed most of the experiments with help from U.F., Z.I., A.N., D.S., R.S.J., and R.M. A.N. performed EMSA under the guidance of P.V.S. Data analysis was performed by B.S. The patient data analysis was performed by A.K.S. and R.S. K.W. and D.N. wrote the manuscript with critical inputs from A.K.S., R.S., M.L.F., and C.A.H. All the authors read and edited the manuscript. Funding and overall supervision of the study was provided by D.N.

## Competing interests

The authors declare no competing interests.
