## [Peer Review File · Nature Communications]

Reviewers' comments:

Reviewer #1 (Remarks to the Author):

This study explores the potential role of a rare SNP that is seen at highest frequency in African populations and associated with a twofold prostate cancer risk as compared to non-SNP carriers. The authors posit that there is a functional role that enables the initiation or development of prostate cancer through alterations in non-coding regions of the genome. Using 4C capture approaches to explore DNA interactions they nominate a set of lncRNAs that are activated and may enhance the activation of MYC, a commonly activated oncogene in all prostate cancer. Validation was performed on the TCGA cohort.

1) The authors are not entirely clear as to what prostate cancer Hi-C data was used for this study. To date there are few published cancer datasets. It would be important to specify the data set(s). There are also data sets using benign prostate cells (RWPE) with and without ERG expression, another important prostate cancer oncogene. Were these data also explored? Addition of these data would be useful to understand how universal these findings are in prostate disease progression. This is particularly of interest because SPDEF is also an ETS transcription factor that was identified.

2) Exploration of the TCGA public data is important to attempt to validate their findings. Despite the small numbers of cases, they do see higher MYC levels in cases with the risk alleles. It would be important to clarify if these cases have or do not have MYC amplifications? This is a common region of genomic amplification in prostate cancer.

3) It is not clear to this reviewer if the outcomes data are valid. The TCGA did not consistently collect outcome data. Is this correct? What are the demographics of these cases used?

4) The authors might extend these validation findings to some of the larger datasets that were generated on advanced prostate cancer (e.g., SU2C-PCF, WES, RNAseq data) or ICGC whole genome data in order to compare the risk allele or other stated associations.

Minor Comments

1) The abbreviation for DNase I hypersensitive sites, DHS, should be defined in the main text. Perhaps even briefly defined for the non-expert.

2) The HiC data is an important inclusion in this study. The authors cite the Wang paper as the main reference, which is providing a convenient browser but is not the original description of Hi-C data.

3) "ESMA experiments". When possible please introduce acronyms to the non-specialist audience.

Reviewer #2 (Remarks to the Author):

This work by Walavalkar et al. describes the regulatory expression of a lncRNA hub modulating a prostate cancer associated enhancer. The authors asked the mechanistic function of a rare snp found in the 8q24 locus associated with greater susceptibility in prostate cancer. Using several CHIP-, ATTAC- and RNA-seq dataset published by different consortiums, they established that the rare variant is located within a region supporting enhancer features. Following cloning the different "natural" variants of this putative enhancer in front of a reporter gene, they show that it embeds transcription activation capacity. The analysis of HiC dataset and 4C single genes experiments indicate that the enhancer region is in close contact with lncRNA promoters associated with cMyc expression.

TCGA clinical and expression dataset suggest that the enhancer variants are associated with poor outcome. Interestingly, the variants define a consensus binding site for the SPDEF androgen transcription factor. Chip experiments show that SPDEF can bind to these rare variants in vivo and support a model in which it further activates the lncRNA expression.

Although the conclusions of the manuscript are very attractive, several major issues need to be addressed for publication in Nature communication, especially much more has to be presented concerning the clinical aspects and enhancer activities.

Major issues/suggestions

1. The luciferase test is poorly explained and has major variabilities that are not explained by the authors. For instance, the comparison between allele G and T without androgen show almost a 2 fold increase in Figure 2B, but none in Figure 2D (green points) and 4 fold change in Figure 5D. These variations strongly compromise the reliability of such measurements. A lentivirus integration might help to get more robust results. In figure 2, it should be presented the vector and the internal normalization process, since it is very complicated to understand for the reader. As it is, it shadows the demonstration the the alleles have a significant differences for the enhancer activity- a key message of the paper. What about chromatin data to support further opening of the region (attac, Mnase etc..) in the different allele conditions?

2. "eRNA expression from the enhancer is not changed". This quite strange since eRNA are supposed to be cryptic RNA. Their role as enhancer is more about the chromatin state and TF binding capacity, rather than the eRNA expression itself. In addition, the eRNA POLII might be activated but the eRNA rapidly degraded. So It would not be a surprise that whatever the enhancer is active or not, the eRNA (already very low) levels remain constant.

3. The comparison with breast tumor TAD (sup S2A) should be properly presented and the figure suppS2A placed in the main part.

4. Figure 4D and 4E are completely unbalanced and cannot be used for the model since the statistics are nowhere near any relevance! 737 cases compared with 15 and 1! The authors should find a better cohort or engage their own cohort if they want to draw any clinical conclusion. I suspect that the absence of the "n" number for Figure 4F would reflect also very poor numbers. If this is the case, it should be removed or placed in supplementary file to be only discussed as preliminary insights. Without good clinical data, the manuscript would miss an important message and could not be accepted in Nature communication.

5 Figure 5B missed 2 important controls: with a scramble oligo and using cellular extract of cells treated with siSPDEF.

6. The experiment in Figure 5C is also a crucial aspect of the manuscript and as it is, it fails to convince the reader if SPDEF binds differently the different alleles. The ChIP analysis has no negative or positive controls to determine whether the SPDEF signals are significant or not. The author should perform the experiment by probing SPDEF known promoters and other negative regions of the genome. Additionally, a normalization gene should be used to confirm that the IP is robust.

7. the competition experiment is very interesting but requires a more drastic dose response. The authors should perform a proper competition by increasing different doses (and measuring it in vivo) of the competitive plasmid/allele, and not just one.

Minor comments/questions

1. figure 1D some tumors have no signal for ATAC seq, is there any reason? Gleasons? Clinical specificity of these tumors?

2. typo line 183 "loos" instead of loops

3. typo line 225 "(Fig. (Fig 4C)"

4. Figure 2C, the bottom of the figure with chip data of FOXA1, DHS or K27ac are not discussed at all. remove

5. In figure 5D, 3Xflag SPDEF should be at the top of the gel.

6. Figure 5G. the legend is missing: is it fold change luciferase? And on the X axis, is it ratio of the plasmids? It should be indicated and the results better described in the text.

We thank reviewers for critical feedback to our manuscript. Please find below the pointwise response to comments/suggestions.

Reviewer #1 (Remarks to the Author):

This study explores the potential role of a rare SNP that is seen at highest frequency in African populations and associated with a twofold prostate cancer risk as compared to non-SNP carriers. The authors posit that there is a functional role that enables the initiation or development of prostate cancer through alterations in non-coding regions of the genome. Using 4C capture approaches to explore to DNA interactions they nominate a set of lncRNAs that are activated and may enhance the activation of MYC, a commonly activated oncogene in all prostate cancer. Validation was performed on the TCGA cohort.

1) The authors are not entirely clear as to what prostate cancer Hi-C data was used for this study. To date there are few published cancer datasets. It would be important to specify the data set(s).

We regret the lack of information on HiC data. The datasets used to plot the LNCaP HiC were GSM2827298/99 from the ENCODE Project Consortium (An integrated encyclopaedia of DNA elements in the human genome. Nature 2012. 489(7414):57-74) and the datasets for T47D were GSM2827515/16, also from ENCODE consortium.

There are also data sets using benign prostate cells (RWPE) with and without ERG expression, another important prostate cancer oncogene. Were these data also explored? Addition of these data would be useful to understand how universal these findings are in prostate disease progression. This is particularly of interest because SPDEF is also an ETS transcription factor that was identified.

We thank the reviewer for pointing out this dataset. We have now plotted the TAD structure at 8q24 locus in RWPE cells with GFP or ERG overexpression (Rickman et al., 2012). Interestingly, 8q24 TAD shows intra-TAD interactions that are less defined in RWPE cells expressing GFP. Notably, the TAD gains more short-range intra-TAD interactions that divides this TAD into less defined sub-TADs upon ERG overexpression. The data suggests that ERG overexpression strengthens the short range interaction and redefines the long range interactions resulting in robust intra-TAD interactions at 8q24 locus. The read depth of HiC, however, did not allow us to annotate the regulatory regions that gain these interactions within the TAD. The gain of short range interactions upon the overexpression of ETS factor is similar to the reported gain in short-range over long-range interactions within the TAD upon estrogen stimulations (Rodriguez et al., 2019, Saravanan et al., 2020). The shift from homogenous large TADs to smaller sub-TADs organization has also been noted in cellular transformation from normal to prostate cancer (Rhie et al., 2019), Importantly in the same study, the gained short range interactions during prostate cancer development exhibited the enrichment of ETS factor motif, corroborating with the findings of Rickman et al., 2012. We have included this data in Fig. S4 and have discussed the data at lines 358-362 in the revised manuscript.

2) Exploration of the TCGA public data is important to attempt to validate their

findings. Despite the small numbers of cases, they do see higher MYC levels in cases with the risk alleles. It would be important to clarify if these cases have or do not have MYC amplifications? This is a common region of genomic amplification in prostate cancer.

We regret for not providing the details in terms of datasets used and the reason for the low sample number. The SNP rs72725854 is rare (6% frequency) and located in the non-coding region. To establish the correlation between the genotype status at SNP and target gene expression, patient samples with both whole genome sequencing (or SNP array) and matched RNA-seq are required. Unfortunately, this SNP and its linked SNPs were not covered in the SNP arrays used in TCGA and other studies. Thus, we analysed the available whole-genome sequencing (WGS) data from the Pan-cancer analysis of whole genome (PCAWG) consortium, consisting of samples from TCGA and ICGC (Campbell et al., Nature 2020). The PCAWG cohort has 2,583 patients with high quality WGS data, out of which only 1,036 have matched RNA seq data (to our knowledge this is the largest cohorts with WGS). From this cohort, we identified 753 samples in total (across all tumor types) with genotype information at SNP locus (737 with A/A, 15 with A/T and one with T/T allele) and also have matched RNA-seq data (as shown in Fig. 4D). Of these, only 19 samples were from Prostate Adenocarcinomas (Fig. 4E). All these 753 samples belonged to the TCGA subset of PCAWG. From the clinical information, we found out that all samples (except one) with A/T or T/T alleles (in Fig. D and E) were from the African ancestry.

In Fig. 4D and 4E, the relative copy number of genes is highlighted with different colors and the legend is shown on the right (where 0 is neutral, 1 is amplification, 2 is high-level amplification, -1 is deletion, -2 is deep deletions). At the pancancer levels, indeed we see that few patients have copy number amplification (Fig. 4D), however, in prostate adenocarcinoma samples, no copy number alteration were observed for myc or the lncRNAs. Thus, the effects seem to be from the overexpression of these genes. We have described these details better in the revised manuscript at lines 237-242.

3) It is not clear to this reviewer if the outcomes data are valid. The TCGA did not consistently collect outcome data. Is this correct? What are the demographics of these cases used?

The number of samples available with whole genome and RNA-seq data is limited (as discussed above). Thus, we asked if patients with any genomic alterations affecting the PCAT1, PVT1 and Myc, independent of the risk allele status and demographic/ancestry, have any effect on the overall survival of the prostate cancer patients?

For this, we exploited the samples available with whole exome/genome/SNP array, and clinical information in cBioportal. This comprises of three main cohorts: Prostate Adenocarcinoma (TCGA, PanCancer Atlas) (n=494), Metastatic Prostate Adenocarcinoma (SU2C/PCF Dream Team, PNAS 2019) (n=444) and Metastatic Prostate Adenocarcinoma (MCTP, Nature 2012) (n=61). Thus, the survival analysis shown in Fig. 4F is not limited to TCGA alone.

Overall, in Fig. 4F, we show that the genomic alterations of PCAT1, PVT1 and MYC (mostly consisting of gene amplification, Fig. S2F) causes poor survival, compared to the patients with no alterations in the above genes independent of the risk allele status and demographic/ancestry.

We have now removed the Fig. 4G which showed the survival analysis based on PVT1 RNA expression data (high vs. low), because these samples have already been covered in Fig. 4F and the gene amplification is known to correlate with the RNA expression.

4) The authors might extend these validation findings to some of the larger datasets that were generated on advanced prostate cancer (e.g., SU2C-PCF, WES, RNAseq data) or ICGC whole genome data in order to compare the risk allele or other stated associations.

We thank the reviewer for the suggestion. As mentioned above in response #2, we mined all the available cohorts such as TCGA, ICGC, and PCAWG for such datasets. However, the data shown in Fig. 4D and 4E is the only data we could obtain from the publicly available whole genome-seq and corresponding RNA-seq. The other cohorts (with WES and SNP-array), regardless of the risk allele status, have been included in the overall survival analysis (Fig. 4F).

Since, the SNP rs72725854 is rare (~6%) and also specific to the African ancestry, very large cohorts are required with genotype and RNA-seq information to test the statistical significance of these expression associations or to draw clinical relevance. Unfortunately, this is going to be a common bottleneck in understanding the functional relevance of all rare variants not only in underrepresented but also in the overrepresented populations/race in the cancer datasets.

Minor Comments

1) The abbreviation for DNase I hypersensitive sites, DHS, should be defined in the main text. Perhaps even briefly defined for the non-expert.

Thank you, we have now described DNase I hypersensitivity sites (DHS) in the main text, lines 113-115.

2) The HiC data is an important inclusion in this study. The authors cite the Wang paper as the main reference, which is providing a convenient browser but is not the original description of Hi-C data.

We regret the lack of information on HiC data. The datasets used to plot the LNCaP HiC were GSM2827298/99 from the ENCODE Project Consortium (An integrated encyclopaedia of DNA elements in the human genome. Nature 2012. 489(7414):57-74) and the datasets for T47D were from GSM2827515/16, also from ENCODE consortium.

3) "EMSA experiments". When possible please introduce acronyms to the non-specialist audience.

We regret the inconvenience. We have included the description of EMSA at line 287-288 in the revised manuscript.

Reviewer #2 (Remarks to the Author):

This work by Walavalkar et al. describes the regulatory expression of a lncRNA hub modulating a prostate cancer associated enhancer. The authors asked the mechanistic function of a rare snp found in the 8q24 locus associated with greater susceptibility in prostate cancer. Using several ChIP-, ATTAC- and RNA-seq dataset published by different consortiums, they established that the rare variant is located within a region supporting enhancer features. Following cloning the different “natural” variants of this putative enhancer in front of a reporter gene, they show that it embeds transcription activation capacity. The analysis of HiC dataset and 4C single genes experiments indicate that the enhancer region is in close contact with lncRNA promoters associated with cMyc expression. TCGA clinical and expression dataset suggest that the enhancer variants are associated with poor outcome. Interestingly, the variants define a consensus binding site for the SPDEF androgen transcription factor. Chip experiments show that SPDEF can bind to these rare variants in vivo and support a model in which it further activates the lncRNA expression. Although the conclusions of the manuscript are very attractive, several major issues need to be addressed for publication in Nature communication, especially much more has to be presented concerning the clinical aspects and enhancer activities.

Major issues/suggestions

1a. The luciferase test is poorly explained and has major variabilities that are not explained by the authors. For instance, the comparison between allele G and T without androgen show almost a 2 fold increase in Figure 2B, but none in Figure 2D (green points) and 4 fold change in Figure 5D. These variations strongly compromise the reliability of such measurements. A lentivirus integration might help to get more robust results. In figure 2, it should be presented the vector and the internal normalization process, since it is very complicated to understand for the reader. As it is, it shadows the demonstration the the alleles have a significant differences for the enhancer activity- a key message of the paper.

We regret the lack of enough details regarding the luciferase assays. We have now added the detailed description in the methods and legends. Briefly, cells were transfected with an equal amount of ‘A’, ‘G’ or ‘T’-luc plasmids along with 3ng of Renilla plasmid for the internal normalizations. The firefly luciferase readings were normalized to Renilla readings and then are plotted as ‘relative values to Renilla’ or, as ‘fold change over ‘A’ or vector alone’ in which case, the Renilla normalized values were further normalized to ‘A’ or empty vector values.

Regarding the variations in reporter readings among the plots, the differences are due to the culture conditions/DHT treatments. Indeed the data in Fig. 2A, 2B and the blue dots in Fig. 2D were obtained under DHT conditions whereas, the data in green dots (Fig. 2D) were obtained without DHT.

For DHT stimulation, the general norm in the field is to grow cells in phenol free media along with charcoal stripped serum (serum is stripped off for all growth factors and hormones) for three days including 16h of vehicle (Methanol) or DHT treatment. Fig. 2B and blue dots of Fig. 2D are identical from same experiments performed under these DHT conditions. Whereas, the green dots in Fig. 2D are under vehicle treatment. In vehicle treated cells, reporter activity of 'A', 'G' and 'T' alleles is almost similar but upon addition of DHT, the T allele but not the A and G show induction suggesting, the higher activity of the T allele in Fig. 2B and 2D (blue dots) is due to androgens.

The Fig. 5E was performed in complete media where growth conditions are favourable due to the presence of basal level of hormones and growth factors thus, the fold changes are much more robust. To test the relative fold changes of A, G and T allele in complete media, we have now repeated the experiments on the cells grown in complete media similar to Fig. 5E. In these conditions, the fold changes in Fig 2B (revised figure) are similar to Fig. 5E (~ 4-fold). We have now replaced original plots with these new datasets (Fig. 2A and B).

We have added these experimental details and the results in the main text at lines 132-133, 141-142, and 149-155.

1b. What about chromatin data to support further opening of the region (attac, Mnase etc..) in the different allele conditions?

We thank the reviewer for the suggestion. we have now performed the DNase I hypersensitivity assays on 'A' and 'T' alleles. Data suggest more open chromatin features on 'T' over 'A' allele. We have included this data as Fig. 5I and have described the result at lines 327-330.

2. "eRNA expression from the enhancer is not changed". This quite strange since eRNA are supposed to be cryptic RNA. Their role as enhancer is more about the chromatin state and TF binding capacity, rather than the eRNA expression itself. In addition, the eRNA POLII might be activated but the eRNA rapidly degraded. So It would not be a surprise that whatever the enhancer is active or not, the eRNA (already very low) levels remain constant.

We and others have previously shown that presence of eRNA at enhancer is the most reliable predictor of functional enhancer activity and that, eRNAs are robustly induced on nuclear receptor-bound enhancer upon estrogen and androgen stimulations (Li et. al, 2013, Wang et al., 2012, Hsieh et al., 2014 etc). These changes in eRNA expression are result of PolIII and transcriptional machinery recruitment.

The reviewer has correctly pointed out that degradation kinetics of eRNAs is faster as compared to mRNAs. In spite of the faster degradation rates, changes in eRNA transcription from ligand responsive enhancers upon ligand addition have been captured by nascent RNA-seq and qRT-PCRs in several studies (Li et al., 2013, Hsieh et al., 2014, Lam et al., 2013, Saravanan et al., 2020). Thus, unaltered eRNAs

levels (Fig. 2E) and unaltered luciferase activity (Fig. 2D) suggest that enhancer with 'A/A' genotype of rs72725854 does not respond to androgen.

Further, eRNAs *per se* have been shown to play a role in target gene expression by recruiting protein machineries relevant to transcription and enhancer promoter looping (Li et al., 2013, Schaukowitch et al., 2014, Pnueli et al. 2015, Hsieh et al., 2014, Zhao et al., 2019). We have described this section with more details in the revised manuscript at lines at 158-162.

3. The comparison with breast tumor TAD (sup S2A) should be properly presented and the figure suppS2A placed in the main part.

We thank the reviewer for suggestion, we have now replotted the comparative interaction matrices of HiC data from T47D and LNCaP at 8q24 region (Fig. 3E). The plot is overlaid by the gene annotations and HiC derived TADs. The enriched interactions between rs72725854/PCAT1 and PVT1/MYC region are seen in LNCaP over T47D (Red pixels).

4. Figure 4D and 4E are completely unbalanced and cannot be used for the model since the statistics are nowhere near any relevance! 737 cases compared with 15 and 1! The authors should find a better cohort or engage their own cohort if they want to draw any clinical conclusion.

We completely agree with the reviewer's concern regarding the low numbers of tumor samples with 'AT' and 'TT' genotype, and for that reason we have not performed any statistical test in Fig. 4D and 4E.

The variant rs72725854 is rare (6% frequency) and located in the non-coding region. To establish the correlation between the genotype status at SNP and target gene expression, we required patient samples with both, whole genome sequencing (or SNP array) and matched RNA-seq data. Unfortunately, this SNP and its linked SNPs were not covered in the SNP arrays used in TCGA and other studies. Thus, we analysed the available whole-genome sequencing (WGS) data from the Pan-cancer analysis of whole genome (PCAWG) consortium, consisting of samples from TCGA and ICGC (Campbell et al., Nature 2020). The PCAWG cohort has 2,583 patients with high quality WGS data, out of which 1,036 have matched RNA seq data as well (to our knowledge this is the largest cohorts with WGS). From this cohort, we identified 753 samples in total (across all tumor types) with genotype information at SNP locus (737 with A/A, 15 with A/T and one with T/T allele) and also have matched RNA-seq data (as shown in Fig. 4D). Of these, only 19 samples were Prostate Adenocarcinoma (Fig. 4E). All these 753 samples belong to the TCGA subset of PCAWG. From the clinical information, we found out that all samples (except one) with A/T or T/T alleles (in Fig. 4D and 4E) are from African ancestry. We have added these details now in the revised manuscript at lines 237-242.

Since, the variant is rare, large cohorts are required with WGS and RNA-seq to test the statistics or to draw clinical relevance. We are afraid that lack of statistics will be a bottleneck for functional genomics involving rare SNP in any race but more so in understudied populations/race.

In spite of these limitations, though without statistics, Fig. 4D and E show the trend of upregulation of the oncogene and lncRNAs in the risk allele carriers. And since, the numbers were obtained from all the possible data sources, it remains the best and only source of data for drawing these associations. Engaging a population of such large numbers for such a variant is beyond the scope of this study.

I suspect that the absence of the “n” number for Figure 4F would reflect also very poor numbers. If this is the case, it should be removed or placed in supplementary file to be only discussed as preliminary insights. Without good clinical data, the manuscript would miss an important message and could not be accepted in Nature communication.

We understand the concern of reviewer. However, rs72725854 being a rare variant, the number of samples available with whole genome DNA and RNA-seq data is limited (as discussed in the above point) thus, they are not suitable for statistical tests. Through functional data including 4C and CRISPRi we show the association of enhancer harbouring rs72725854 with higher expression of PCAT1, PVT1 and myc (Fig. 3 and 4C). Hence, we asked if patients with any genomic alterations affecting the PCAT1, PVT1 and Myc, independent of the risk allele status and demographic/ancestry, have any effect on the overall survival of the prostate cancer patients. For this, we exploited the samples available with whole exome/genome/SNP arrays, and clinical information in cBioportal. This comprises of three main cohorts: Prostate Adenocarcinoma (TCGA, PanCancer Atlas) (n=494), Metastatic Prostate Adenocarcinoma (SU2C/PCF Dream Team, PNAS 2019) (n=444) and Metastatic Prostate Adenocarcinoma (MCTP, Nature 2012) (n=61). Thus, the survival analysis shown in Fig. 4F is not limited to TCGA alone.

Overall, in Fig. 4F, we show that independent of the risk allele status and demographic/ancestry, the genomic alterations of PCAT1, PVT1 and MYC (which are mostly amplification, Fig. S2F) cause poor survival, compared to the patients with no alterations in the above genes.

To summarise, for the risk allele, all possible datasets available till date were checked and this is a general problem for the study of any such rare variation. The study in-fact tries to unravel how such rare variations might be implicated in disease susceptibilities since it is almost impossible to draw conclusions for such cases from purely clinical studies due to absolute lack of numbers, even more though for a rare variant found in a particular demographic, for which lack of such data is more acute.

5 Figure 5B missed 2 important controls: with a scramble oligo and using cellular extract of cells treated with siSPDEF. The experiment in Figure 5C is also a crucial aspect of the manuscript and as it is, it fails to convince the reader if SPDEF binds differently the different alleles.

We thank the reviewer for suggesting these important controls. We have now included the EMSA results with scrambled oligos (Fig. S3B) and siSPDEF (Fig. S3C). The scrambled oligos with similar GC content as that of ‘A’ and ‘T’ oligos, failed to bind with SPDEF and showed a binding pattern similar to the ‘A’ allele (Fig. S3B). We performed siRNA mediated knockdown of SPDEF in LNCaP cells to

compare the complex bound on 'T' oligos with that of scr siRNAs. The complex bound on 'T' allele appears weaker upon SPDEF knockdown (Fig. S3C) conforming, the presence of SPDEF in this complex. We have described these new results at lines 291-294.

The EMSA results recapitulate the loss of transcriptional activity of 'T' allele but not 'A' allele upon SPDEF knockdown (Fig. 5E). Further, the loss of DHT response by 'T' allele upon SPDEF knockdown (Fig. 6F) also confirms the dependency of 'T' allele on SPDEF for its active regulatory potential. Higher regulatory potential of 'T' allele is also reflected from its higher reporter activity (Fig. 2B) and its more open chromatin features over 'A' allele (Fig. 5I). Together, these results show the risk allele 'T' binds with SPDEF which is required for its gained transcriptional response to DHT.

The ChIP analysis has no negative or positive controls to determine whether the SPDEF signals are significant or not. The author should perform the experiment by probing SPDEF known promoters and other negative regions of the genome. Additionally, a normalization gene should be used to confirm that the IP is robust.

As per reviewer's suggestion, we have now compared the binding of SPDEF on rs72725854 alleles with the known SPDEF-bound regions and also the potential non-bound regions based on SPDEF ChIP-seq in VCaP cells (Wei et al., 2010). We performed ChIP-qPCRs on the PSA promoter which is known to bind with SPDEF (Oettgen et al., 2000), it showed ~80-fold enrichment over beads control when normalized with original input chromatin. Similarly, another SPDEF-bound region, PCAT1 intron exhibited ~40-fold enrichment over beads again after normalizing with input material. The rs72725854 with 'T' allele exhibited a ~30-fold enrichment. The binding on 'A' and 'G' alleles was slightly more or equivalent to the two negative controls taken in the study (Fig. 5C). These positive and negative controls were verified from the SPDEF ChIP-seq in VCaP cells (Wei et al., 2010). We have described this new data at lines 294-304.

7. the competition experiment is very interesting but requires a more drastic dose response. The authors should perform a proper competition by increasing different doses (and measuring it in vivo) of the competitive plasmid/allele, and not just one.

As suggested by the reviewer, we have increased the number of doses of 'A' and 'T' -non luc alleles over 'T' allele-Luc in competition assay. We observe a dose dependent loss in reporter activity of 'T'-allele luc upon increasing doses of 'T'-non luc (Fig. 5H, blue bars). Though, the increasing doses of 'A'-non luc also exhibited some competition however, the pattern of loss was stochastic even with the highest dose of 'A'-non luc (Fig. 5H, red bars). These effects could be due to the weak binding events of SPDEF on 'A'-non as oppose to the dose dependent strong binding of SPDEF on 'T'-non luc. The data suggest that the 'T' allele is able to better compete for SPDEF as it exhibits stronger binding ability (Fig. 5H). We have now described this section with more details in the main text at lines 318-327 and in methods at lines 462-471.

Minor comments/questions

1. figure 1D some tumors have no signal for ATAC seq, is there any reason? Gleasons? Clinical specificity of these tumors?

Some samples have less signal at the rs72725854 enhancer region compared to other tumors, which possibly represents heterogeneity across different patients. Though, the relative signal is still higher at sites where ATAC signal is higher in other tumors. Unfortunately, Gleason score for these tumors was not available.

2. typo line 183 “loos” instead of loops

We thank reviewer for pointing it out.

3. typo line 225 “(Fig. (Fig 4C)”

We thank reviewer and have corrected the label in the revised manuscript.

4. Figure 2C, the bottom of the figure with chip data of FOXA1, DHS or K27ac are not discussed at all. Remove

We thank reviewer for pointing out this oversight. The data is removed in the edited Fig. 2C.

5. In figure 5D, 3Xflag SPDEF should be at the top of the gel.

We have now improved the labelling of immuno-blot panels in Fig. 5D. We have used +/- signs against scr, siSPDEF and 3xFlag SPDEF overexpressions.

6. Figure 5G. the legend is missing: is it fold change luciferase? And on the X axis, is it ratio of the plasmids? It should be indicated and the results better described in the text.

We have now improved the legends and have indicated them better in the revised Fig. 5G and in the methods. The y-axis is fold luciferase change over 100% A allele and the x-axis is the ratio of T/A plasmids ranging from 0, 20, 40, 60, 80 and 100%.

REVIEWERS' COMMENTS:

Reviewer #1 (Remarks to the Author):

The authors have addressed my questions.

Reviewer #3 (Reviewer Replacement to comment on behalf of Reviewer #2, Remarks to the Author):

The authors have provided thorough responses to the reviewer comments.